# Resolving atomic SAPO-34/18 intergrowth architectures for methanol conversion by identifying light atoms and bonds

Boyuan Shen [1], Xiao Chen [1✉], Xiaoyu Fan [1], Hao Xiong [1], Huiqiu Wang [1], Weizhong Qian[1], Yao Wang [1✉] & Fei Wei [1✉]

The micro-structures of catalyst materials basically affect their macro-architectures and catalytic performances. Atomically resolving the micro-structures of zeolite catalysts, which have been widely used in the methanol conversion, will bring us a deeper insight into their structure-property correlations. However, it is still challenging for the atomic imaging of silicoaluminophosphate zeolites by electron microscopy due to the limits of their electron beam sensitivity. Here, we achieve the real-space imaging of the atomic lattices in SAPO-34 and SAPO-18 zeolites, including the Al–O–P atoms and bonds, by the integrated differential phase contrast scanning transmission electron microscopy (iDPC-STEM). The spatial distribution of SAPO-34 and SAPO-18 domains in SAPO-34/18 intergrowths can be clearly resolved. By changing the Si contents and templates in feed, we obtain two SAPO-34/18 catalysts, hierarchical and sandwich catalysts, with highly-mixed and separated SAPO-34 and SAPO-18 lattices respectively. The reduced diffusion distances of inside products greatly improve the catalytic performances of two catalysts in methanol conversion. Based on the observed distributions of lattices and elements in these catalysts, we can have a preliminary understanding on the correlation between the synthesis conditions and structures of SAPO-34/18 intergrowth catalysts to further modify their performances based on unique architectures.

[1] Beijing Key Laboratory of Green Chemical Reaction Engineering and Technology, Department of Chemical Engineering, Tsinghua University, Beijing, China. ✉email: chenx123@tsinghua.edu.cn; wang_yao@tsinghua.edu.cn; wf-dce@tsinghua.edu.cn

The efficient utilization and conversion of energy boosted the development of human civilization. Considering the limit of fossil energy reserves and the energy structures in East Asia, it is of great importance to explore a new coal chemical industry represented by the methanol conversion[1–4]. Following the concept of the "methanol economy," the methanol conversion over various porous zeolite catalysts has been rapidly developed and gradually started the pathway on the scale-up process in the past several decades[3–10]. In the relevant studies on heterogeneous catalysis, the modification of catalyst structures is a core means to improve the diffusion efficiency and the catalytic performances. For example, some classical strategies for the nano-sized, lamellar, and hierarchical catalyst structures have been reported to work[11–21]. It is expected to directly obtain the target macroarchitectures of catalysts by modifying their microstructures. Meanwhile, due to the complex channel structures in zeolites, many typical conversion processes over zeolite catalysts will follow the mechanism of double cycles in hydrocarbon pools[22–28]. In both the olefin and aromatic cycles, the confinement effect of the zeolite framework on the hydrocarbon pool species plays a most important role in the product distribution, diffusion, and catalyst lifetime during the methanol conversion.

Therefore, solid characterizations of catalyst structures and hydrocarbon pool species are highly desired to reveal the structure–property correlation and reaction mechanism. Electron microscopy is an efficient tool to resolve the atomic structures with a high spatial resolution[29–31]. However, for these porous materials, the inevitable damage by high-energy electrons and the low contrasts of light elements will make it challenging to obtain high-quality images in traditional imaging modes. Fortunately, the integrated differential-phase contrast scanning transmission electron microscopy (iDPC-STEM)[32–38] has been proven to be powerful for the imaging of electron-sensitive materials with light elements[39–44], such as the zeolites and metal-organic frameworks (MOFs), which provides new possibilities for us to resolve the porous catalysts and inside hydrocarbon pools in real space.

SAPO-34 and SAPO-18 are two kinds of zeolite catalysts that were most widely used in the methanol-to-olefins (MTO) process. SAPO-34 is a silicoaluminophosphate zeolite with a CHA-type framework[45,46]. It is composed of the molecular cages connected by cross-linked channels with a cage size of 7.4 Å and a channel size of only 3.8 Å. SAPO-18 is another silicoaluminophosphate zeolite with an AEI-type framework[47,48]. Its cage and channel structures are very similar to those of SAPO-34 with nearly the same sizes. SAPO-34 and SAPO-18 lattices can perfectly match in one direction and usually form intergrowth structures by using the same structure-directing agents as templates, such as triethylamine (TEA) and tetraethylammonium hydroxide (TEAOH)[49–53]. The cages and channels in SAPO-34 and SAO-18 not only allow the diffusion of ethylenes and propylenes but also prevent the diffusion of aromatics and higher-molecular-weight products. Therefore, SAPO-34 and SAPO-18 can be used as zeolite catalysts with high selectivity of ethylenes and propylenes in the MTO process together. Meanwhile, due to the large cages inside, aromatics and heavy hydrocarbons will be generated in these cages but cannot diffuse through channels, which will induce coking and deactivation of zeolite catalysts by changing the channel connectivity.

In the SAPO-34/18 intergrowth, the complex distribution of different domains and the infinite stacking sequences determine the diverse microstructures during the assembly of CHA and AEI frameworks, and further affect their macroarchitectures for catalysis. However, to date, the atomic structure information of SAPO-34 and SAPO-18 zeolites have not been obtained from real-space imaging. Therefore, this traditional industry is facing a major bottleneck that we cannot fundamentally reveal the correlation between catalytic performances and atomic local structures. We believe that the progresses in imaging techniques will provide us the atomic information of these zeolite catalysts and guide the synthesis and modification of target structures for tailored functions.

In this work, we use the iDPC-STEM to achieve the atomic imaging of SAPO-34 and SAPO-18 lattices. Based on these observations, we atomically resolve the SAPO-34/18 intergrowth architectures for the MTO catalysis. The specific distribution and sequence of SAPO-34 and SAPO-18 domains in these catalysts are revealed by the direct imaging. Then, combining the element analysis and electron microscopy, we can understand the origin of various microstructures based on the effect of Si content and templates, which further induces the evolution of different macroarchitectures. Finally, we investigate the catalytic performances of different SAPO-34/18 intergrowth catalysts. These results help us establish a clear structure–property correlation in zeolite catalysts and provide a solid basis for the further modification of catalyst structures.

## Results and discussion

**SAPO-34/18 intergrowth architectures**. The traditional SAPO-34/18 intergrowth usually forms the nearly cuboid particles with micrometer sizes as shown in Fig. 1a. These samples were synthesized using a pure TEA template under the conditions of high Si content (Si/Al ratio of 0.32 in feed). Due to their large crystal sizes, the lifetime and selectivity of catalysts are limited by the long diffusion distances. Therefore, the modification of catalyst structures is required. First, after reducing the Si content (Si/Al ratio of 0.11 in feed), there are mesoporous and macroporous structures constructed inside these micron-sized solid cuboid particles as shown in Fig. 1b and Supplementary Figures 1 and 2. The volume of these mesopores and macropores can be reduced by increasing the crystallization temperature and time to form nearly solid particles again. Both the mesopores and macropores contribute to the pore structures in this sample, so it is named as the hierarchical SAPO-34/18 catalyst. Then, we successfully changed the catalyst shape from cuboid particles to nano-thick lamellas using the dual template (TEAOH and TEA) method. Figure 1c and Supplementary Figures 3 and 4 reveal the structure of the so-called sandwich SAPO-34/18 catalyst, where the lamellas are stacked like a sandwich. These two modified SAPO-34/18 intergrowth catalysts both have abundant mesopores and macropores due to the hierarchical and lamellar architectures, which cannot be observed in the traditional cuboid catalysts (Supplementary Figure 5). They will lead to shorter diffusion distances for guest molecules in at least one dimension and then efficiently accelerate the diffusion of these molecules in bulk catalysts and increase the reaction contact areas.

However, since there is a lack of characterization and understanding of microstructures in these catalysts, we still cannot reveal the correlation between catalyst architectures and synthesis conditions. These attempts, based on the synthesis experience, cannot guide the follow-up modification of catalyst structures and performances fundamentally. It is required to understand the SAPO-34/18 intergrowth from the atomic scale. To simplify the analysis of SAPO-34/18 catalyst structures, we introduce a cuboid frame in Fig. 1d to describe three nearly orthogonal directions of this cuboid by the letters α, β, and γ. The corresponding crystallographic axes of SAPO-34 (CHA) and SAPO-18 (AEI) are marked in Fig. 1d. In the following discussions, we use the directions α, β, and γ to describe the crystallographic axes of SAPO-34/18 intergrowth concisely.

Then, we obtain a deeper insight into the atomic modelling of SAPO-34/18 intergrowth. As shown in Fig. 1e and Supplementary

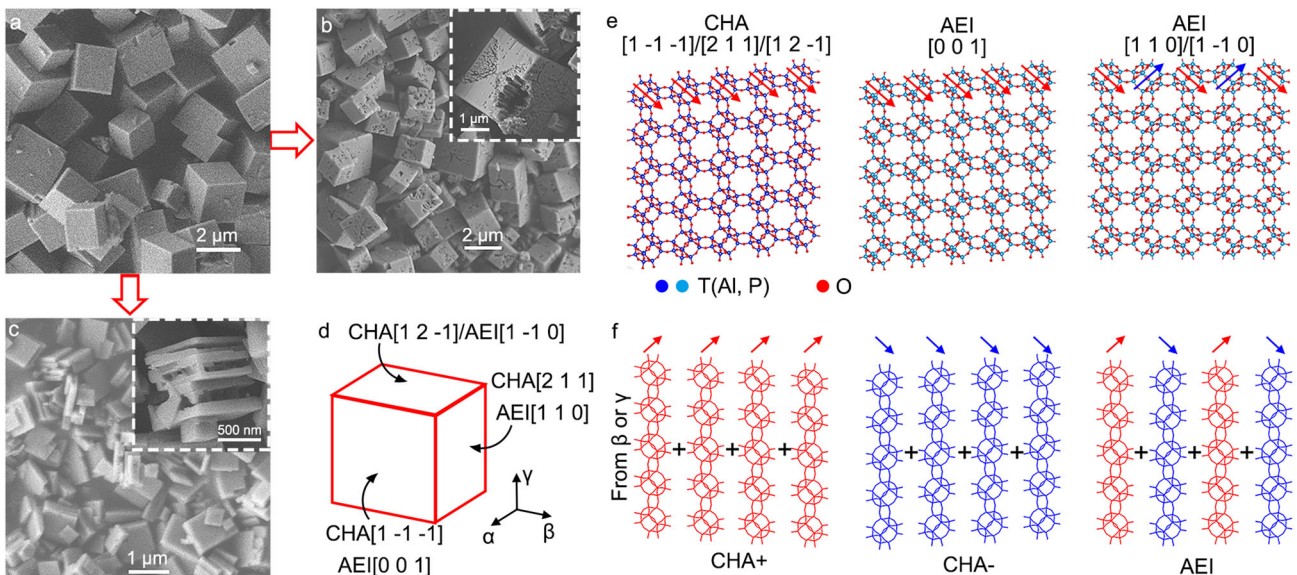

**Fig. 1 SAPO-34/18 intergrowth catalysts. a** SEM image showing the traditional micron-sized cuboid architectures of SAPO-34/18 catalysts. **b, c** SEM images of the modified SAPO-34/18 intergrowth catalysts with hierarchical (**b**) and sandwich (**c**) architectures. **d** Cuboid model showing the crystallographic axes of SAPO-34 (CHA) and SAPO-18 (AEI) in the intergrowth crystals. **e** Structural model of the CHA and AEI frameworks viewed from the different crystallographic axes given in (**d**). **f** Layer-based model of CHA and AEI frameworks viewed from direction β or γ.

Figure 6, the [0 0 1] projection of SAPO-18 (AEI) highly matches with three principle projections of SAPO-34 (CHA). Therefore, the lattices of SAPO-34 and SAPO-18 can be connected from the [0 0 1] direction of AEI (direction α in Fig. 1d) with negligible distortion. From the direction β or γ, we can study the SAPO-34/18 structures with a layer-based model[49] given in Fig. 1f, where the CHA and AEI lattices can be distinguished by AA or AB stacking of unit layers. There are two types of unit layers with a mirror symmetry as marked by the red and blue arrows, respectively. Random connection of these layers will generate infinite stacking sequences of SAPO-34/18 as fingerprint information of each intergrowth crystal, which can only be studied directly by real-space imaging.

**Atomic imaging of SAPO-34 and SAPO-18 lattices.** For a long time in the past, we have been short of high-resolution imaging methods for zeolites until the iDPC-STEM improved the imaging ability of various beam-sensitive materials to the atomic level. Figure 2a shows the iDPC-STEM image of SAPO-34/18 intergrowth from direction β or γ. The connection between SAPO-34 (CHA) and SAPO-18 (AEI) domains is clearly observed in this image, consistent with the structure modelling in Fig. 1. In the fast Fourier transform (FFT) pattern (Fig. 2b), high-frequency reflections are displayed with an information transfer of 0.94 Å. Then, the magnified image of SAPO-18 (AEI) domain in Fig. 2c further helps us to resolve the atomic structures that perfectly match the atomic model. Since the atomic contrasts in iDPC-STEM image are nearly linear with the atomic numbers, the Al, O, and P atom columns can be identified by their different contrasts in the profile analysis. The bottom panel in Fig. 2c gives the intensity profile of the area marked by the red frame in the iDPC-STEM image. Based on three intensity peaks of different elements in the profile, the projected positions of Al, O, and P columns are indicated. It is worth mentioning that a trace of Si doping will not influence the contrasts (relative peak heights) of Al and P atoms during the element identification.

Meanwhile, the projected positions of Al, O, and P columns reveal the different bond lengths of Al–O and P–O bonds. Theoretically, the Al–O bonds are 0.2 Å longer than the P–O

bonds. In the intensity profile in Fig. 2c, the O column is clearly closer to the P column, which indicates that these P–O bonds are really shorter from this projection. Similar atomic information can also be obtained from the SAPO-34 (CHA) domain. By comparing the two images in Fig. 2c, d, we can better understand the AA and AB stacking of the unit layers by the O bridges in SAPO-18 (AEI) and SAPO-34 (CHA), respectively. In Fig. 2d, the profile analysis of the area in the blue frame in the iDPC-STEM image also shows different contrasts of Al, O, and P columns and the different projected lengths of Al–O and P–O bonds. These results provide us new possibilities to investigate the intergrowth and stacking of zeolite frameworks from the perspectives of atoms and bonds.

Moreover, Fig. 2e provides the atomic connection of the SAPO-34 (CHA) and SAPO-18 (AEI) domains from another projection that is corresponding to the AEI [1 0 0] and CHA [1 1 0] direction. The FFT pattern in Fig. 2f shows an information transfer of 1.10 Å. In the magnified images (Fig. 2g, h), the comparison of the images and models in both SAPO-34 (CHA) and SAPO-18 (AEI) domains reveals the positions of atom columns in the framework. In the profile analysis in Fig. 2g, h, three atom columns are identified by the intensity peaks. However, from this projection, the Al/P columns are composed of both Al and P atoms alternately. Therefore, the intensity peaks of Al/P columns are of nearly equal height, and the average position of O atoms is right in the middle of two Al/P columns. All these observations are consistent with the results that we predicted based on the structure modelling.

**Resolving the microstructures in different SAPO-34/18 catalysts.** Combining the atomic imaging with powder X-ray diffraction (PXRD) results, we can further reveal the local microstructures of specific SAPO-34/18 intergrowth catalysts, especially the stacking sequences of SAPO-34/18. For the traditional cuboid catalysts, although their atomic structures cannot be obtained due to their large sizes against thin sample approximation, the PXRD results in Fig. 3a provide some structural information according to the characteristic AEI peaks at 2θ of 10.7° and 17.0° (corresponding to the AEI (111), (022), and (113) planes) in Fig. 3b, c. We found that

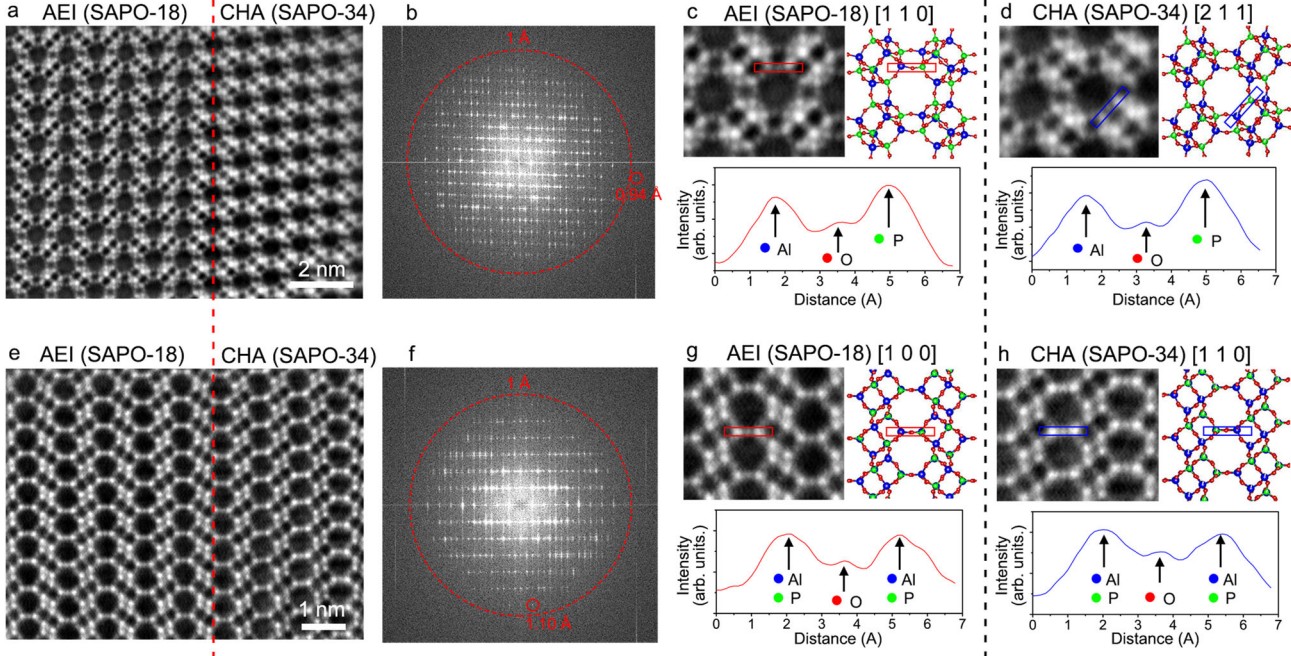

**Fig. 2 Atomic imaging of SAPO-34 and SAPO-18 lattices. a** iDPC-STEM image of SAPO-34 and SAPO-18 lattices from the AEI [110] projection. **b** FFT pattern of **a** showing an information transfer of 0.94 Å. **c, d** Magnified images and intensity profiles of SAPO-18 (**c**) and SAPO-34 (**d**) in (**a**), where the Al, O, and P atoms can be clearly identified. **e** iDPC-STEM image of SAPO-34 and SAPO-18 lattices from the AEI [100] projection. **f** FFT pattern of (**e**) showing an information transfer of 1.10 Å. **g, h** Magnified images and intensity profiles of SAPO-18 (**g**) and SAPO-34 (**h**) in (**e**).

the SAPO-34 is dominant in the cuboid catalysts, since there is no peak at $2\theta$ of 10.7° (Fig. 3b) and the peak at $2\theta$ of 17.0° show a much lower relative height compared with side peaks than that in the hierarchical and sandwich catalysts (Fig. 3c, indicating a few SAPO-18 faults). The hierarchical and sandwich catalysts, however, consist of a rich mixture of SAPO-34 and SAPO-18 lattices. Fortunately, local thin areas that are proper for STEM imaging can be found inside the hierarchical and sandwich SAPO-34/18 crystals from the direction β or γ.

Figure 3d shows the HAADF-STEM image of a hierarchical SAPO-34/18 crystal from the direction β or γ. The dark holes inside the solid framework represent through mesopores, macropores, and hollow areas. Then, the iDPC-STEM was used to resolve the atomic local structures in a thin area of the solid framework in Fig. 3e. The red and blue arrows above the image can point out whether adjacent unit layers are AA or AB stacked. Repeated sequences of AAB, ABA, and ABB indicate that the SAPO-34 and SAPO-18 lattices are highly mixed in the hierarchical catalysts without definable domains and interfaces (more images in Supplementary Figure 7). It is consistent with the PXRD result of the hierarchical catalysts in Fig. 3a–c. The peak at $2\theta$ of 10.7° is very wide and nearly invisible since the sizes of SAPO-18 domains are too small (no definable domains), while the peak at $2\theta$ of 17.0° is quite obvious.

We further compared the stacking sequences at two sides of small mesopores (<10 nm), which may help us know how these small mesopores were generated. In both SAPO-34 and SAPO-18, the stacking sequence is usually consistent in a single continuous unit layer, no matter how large it grows finally. However, as shown in Fig. 3f, g and Supplementary Figure 8, the stacking sequences at the top and bottom of these small mesopores are always different, where the black and white dashed lines indicate the matched and mismatched lattices of the same layers respectively. This means that the growth of new layers in large crystals may start from different sites of surface layers, and then

these separated fragments gradually connect to form a complete layer. For the small mesopores (<10 nm) in Figs. 3f, g and Supplementary Figure 8, the frequent switches between AA and AB stacking (highly mixed SAPO-34 and SAPO-18 lattices) make the lattice mismatch in the same layer more likely to exist than that in single domain, and then cracks are generated as small mesopores. For the large ones (>10 nm), they cannot be simply explained by the lattice mismatching. It is more likely that different growth rates between SAPO-34 and SAPO-18 along βγ plane will slow down the overall crystallization of the hierarchical catalysts so that the separated parts cannot link in a limited time and just form large mesopores, macropores, and even larger hollow areas. As we mentioned above, most of these large pores can be healed to form nearly solid particles again after increasing the temperature and time of crystallization, while small pores caused by lattice mismatch will always be there.

However, the spatial distribution of SAPO-34 and SAPO-18 domains is totally different in the sandwich SAPO-34/18 catalysts. In Fig. 3i–k, we imaged the stacking sequences of a sandwich SAPO-34–18 crystal. These iDPC-STEM images were obtained from the areas as marked by the colored dashed line frames in Fig. 3h. First, Fig. 3i provides a combination of three images showing all the stacking sequences sandwiched between two lamellas. Then, the lattices of the nano-thick lamellas are given in Fig. 3j, k. In these images, we find that SAPO-34 and SAPO-18 domains are completely separated. The SAPO-34 domains on both sides grow faster to form the large lamellas, while the SAPO-18 domain only forms the joint of lamellas with few SAPO-34 faults inside. The domain size of SAPO-34 is limited by the mixing and dispersing of dual templates, which fundamentally determines the thickness of lamellas. Based on these observations, we realize that the special stacking modes and sequences of the unit layers will induce rich local structures of SAPO-34/18 intergrowth catalysts, such as the hierarchical pores and the lamellas in this work.

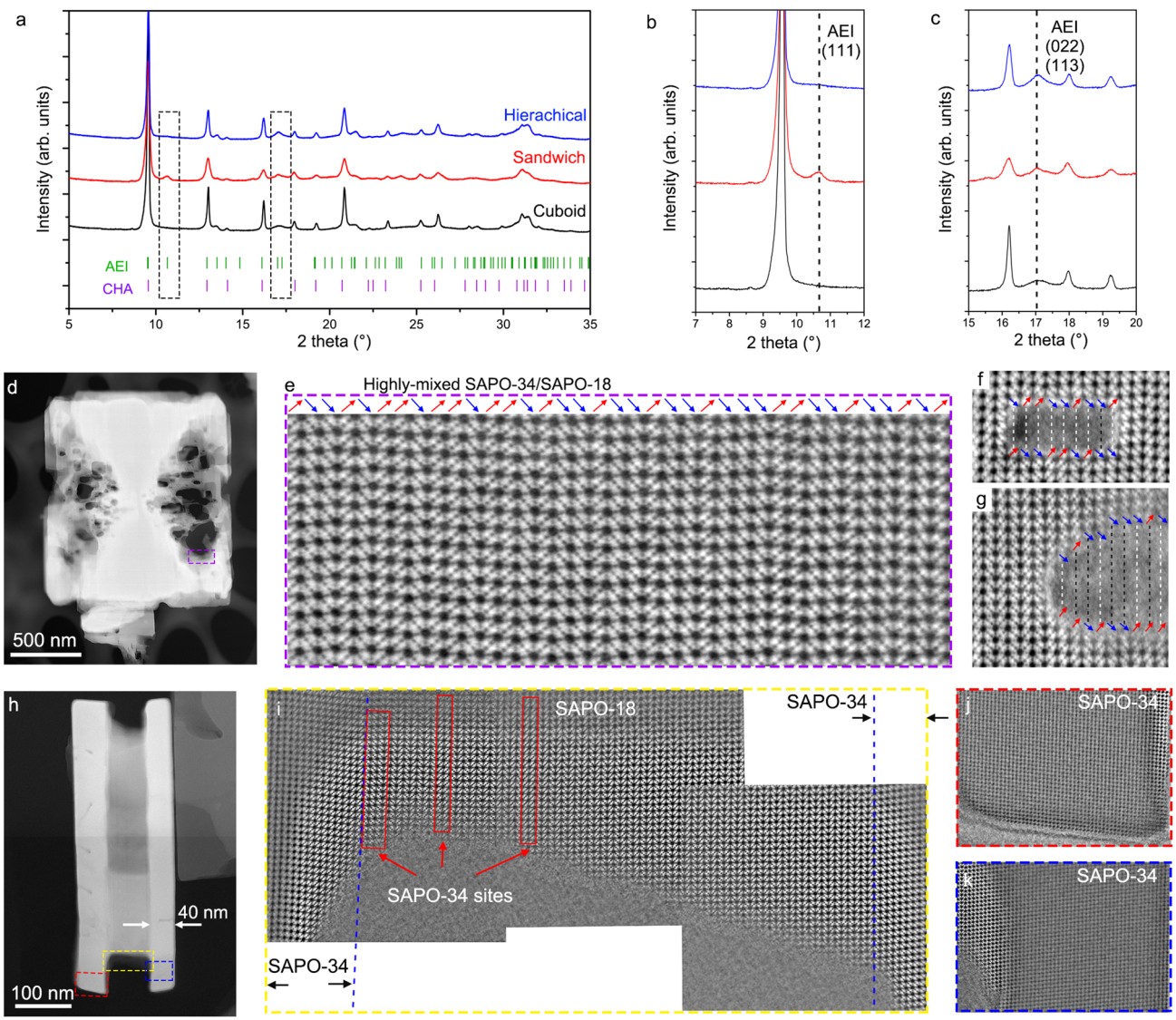

**Fig. 3 Resolving different SAPO-34/18 intergrowth catalysts. a–c** PXRD results of three SAPO-34/18 intergrowth catalysts. The blue, red, and black lines represent the PXRD results of hierarchical, sandwich, and micron-sized cuboid samples, respectively. **d** HAADF-STEM image of a hierarchical SAPO-34/18 crystal from direction β or γ. **e** iDPC-STEM image of the area marked by the dashed line frame in (**d**), showing the highly mixed SAPO-34 and SAPO-18 lattices inside. **f, g** iDPC-STEM images showing the stacking sequences surrounding the through mesopores in the hierarchical SAPO-34/18 catalysts. **h** HAADF-STEM image of a sandwich SAPO-34/18 crystal from direction β or γ. **i** Combination of three iDPC-STEM images showing all the stacking sequences sandwiched between two lamellas (the area marked by the yellow dashed line frame in **h**). **j, k** iDPC-STEM images showing the SAPO-34 lattices of the lamellas on both sides (the areas marked by the red and blue dashed line frames in **h**).

**Controlling SAPO-34/18 intergrowth by Si contents and templates**. The iDPC-STEM imaging reveals the atomic local structures of these two SAPO-34/18 intergrowth catalysts. These results provide us new possibilities to bridge the synthesis conditions, microstructures, and macroarchitectures of SAPO-34/18 catalysts. Based on the experiences on zeolite synthesis, we confirmed two important laws on structural modification of SAPO-34/18 intergrowth by a variable-controlling method. First, it has been reported in many relevant works that higher Si content in the initial gel mixture will preferentially induce the formation of SAPO-34, while lower Si content is favorable for SAPO-18[14,50]. Using the dual templates in this work (see "Methods"), we indeed observed such selectivity of SAPO-34 and SAPO-18 depending on different Si contents in feed in Fig. 4a. According to the changes of characteristic AEI peaks at 2θ of 10.7° and 17.0°, it can be identified that with the increase of Si content, the content of SAPO-18 domains decreases

gradually, and there seems to be only SAPO-34 at the Si/Al ratio of 0.15 in feed.

Then, the content ratio of dual templates also affects the selectivity of products, where pure TEAOH is usually used to direct zeolite structures to SAPO-34 and pure TEA will induce SAPO-18 domain (forming SAPO-34/18 intergrowth)[51,52]. It is also confirmed by a series of syntheses with a fixed Si content as shown in Fig. 4b and Supplementary Figure 9 (see "Methods"). At high TEAOH content (the TEA/TEAOH ratios of 0 and 0.29), we can hardly observe the characteristic AEI peaks at 2θ of 10.7° and 17.0°, where SAPO-34 domains are dominant and SAPO-18 lattices are treated as stacking faults. However, with TEA increased, these two peaks can be observed and the increase of peak heights indicates that there are more SAPO-18 lattices in the intergrowths. Meanwhile, although the Si content in feed is the same, the real Si/Al ratio in TEAOH-template framework (0.16, measured by the X-ray fluorescence (XRF) analysis) is higher

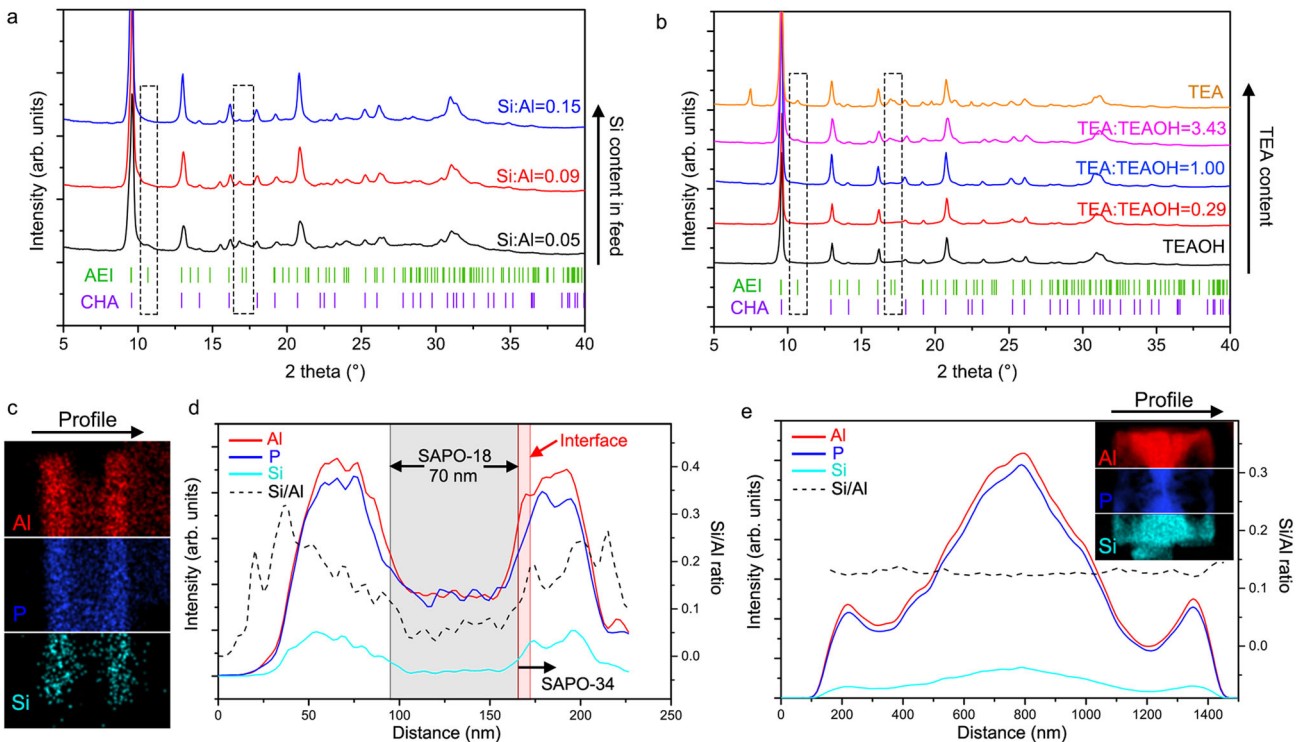

**Fig. 4 SAPO-34/18 intergrowth controlled by Si contents and templates. a** PXRD results of SAPO-34/18 intergrowth synthesized with different Si contents in feed. The blue, red, and black lines represent the PXRD results of SAPO-34/18 samples with Si/Al ratios in feed of 0.15, 0.09, and 0.05, respectively. **b** PXRD results of SAPO-34/18 intergrowth directed by different combinations of TEAOH and TEA dual templates. The yellow, pink, blue, red, and black lines represent the PXRD results of SAPO-34/18 samples with TEA/TEAOH ratios of ∞, 3.43, 1.00, 0.29, and 0 respectively. **c** EDS mapping of the sandwich SAPO-34/18 crystal in Fig. 3h. **d** Elemental profile analysis showing the distribution of Al, P, and Si elements (the red, blue, and cyan lines) and the change of Si/Al ratio (the black dashed line) along the profile direction. **e** Elemental profile analysis and the EDS mapping of the hierarchical SAPO-34/18 crystal in Fig. 3d.

than that in TEA-template framework (0.11). These results can be interpreted as the fact that the cationic groups in TEAOH show a stronger affinity to the negative Si–O–Al units of inorganic segments in the initial gel mixture, and more Si acid sites are integrated into the bulk frameworks.

Therefore, the distribution of Si element in two SAPO-34/18 intergrowth catalysts may determine the selectivity of AA/AB stacking and the spatial distribution of SAPO-34 and SAPO-18 domains. Figure 4c shows the mapping of three elements in the sandwich SAPO-34/18 crystal that we imaged in Fig. 3h using the energy-dispersive spectroscopy (EDS). The elemental profiles of the whole crystal are given in Fig. 4d. Based on the profile analysis, the Si content of SAPO-34 domain is significantly higher than that of SAPO-18 domain, since the Si/Al ratio rises from 0.05–0.1 (for SAPO-18 domain) to 0.15–0.2 (for SAPO-34 domain) through the domain interface (Supplementary Figure 10 and Supplementary Table 1). On the contrary, the elemental profiles of a hierarchical SAPO-34/18 crystal in Fig. 4e indicate that the Si/Al ratios in the highly mixed SAPO-34/18 intergrowth maintain constant in different areas (Supplementary Figure 11 and Supplementary Table 2). There is no Si enrichment in the hierarchical crystal due to the undefinable SAPO-34 or SAPO-18 domain. This comparison reveals that the spatial distribution of SAPO-34 and SAPO-18 domains is closely related to that of Si element, which also helps us realize the probable correlation between structures and synthesis conditions.

In the synthesis of the hierarchical SAPO-34/18 catalyst, we found a balanced value of Si content in feed (Si/Al ratio = 0.11) that makes the TEA template induce the AB and AA stacking of unit layers with similar probabilities. As a result, the SAPO-34

and SAPO-18 lattices are highly mixed in these crystals, which, as we mentioned above, is related to the formation of the hierarchical pores and hollow areas. Meanwhile, in the synthesis of the sandwich catalyst, the TEAOH and TEA templates induce the synthesis of SAPO-34 and SAPO-18 domains, respectively, where the SAPO-34 domains form the lamellas linked by the SAPO-18 domain. Owing to new imaging techniques, we can preliminarily reveal how we obtain these different catalyst structures by simply changing some synthesis conditions, such as the Si contents and templates.

**Catalytic performances of SAPO-34/18 catalysts.** In order to illustrate the influence of crystal size on the lifetime of catalyst, we further examined the catalytic performances of three SAPO-34/18 catalysts (the hierarchical, sandwich, and micron-sized cuboid catalysts) in the methanol conversion. To exclude the effect of acid density, we used three catalysts with nearly the same Si/Al ratios in frameworks (~0.15, measured by the XRF). The hierarchical and sandwich catalysts are the samples that we have studied in Figs. 1–3. The micron-sized cuboid catalysts were synthesized at a higher temperature (see "Methods") to heal the hollow areas to form the nearly solid particles with a high growth rate (Supplementary Figure 12). The catalytic tests were carried out at 723 K with weight hourly space velocity (WHSV) of $4.0\,h^{-1}$ in a fixed-bed reactor (see "Methods"). The catalytic performances of three SAPO-34/18 catalysts are given in Fig. 5a, b and Supplementary Figure 13. All three catalysts show near 100% conversion rate of methanol in the early stage of the reaction. However, after 90 min, the conversion rate of methanol over micron-sized cuboid catalyst decreased rapidly, while the other two catalysts could maintain a

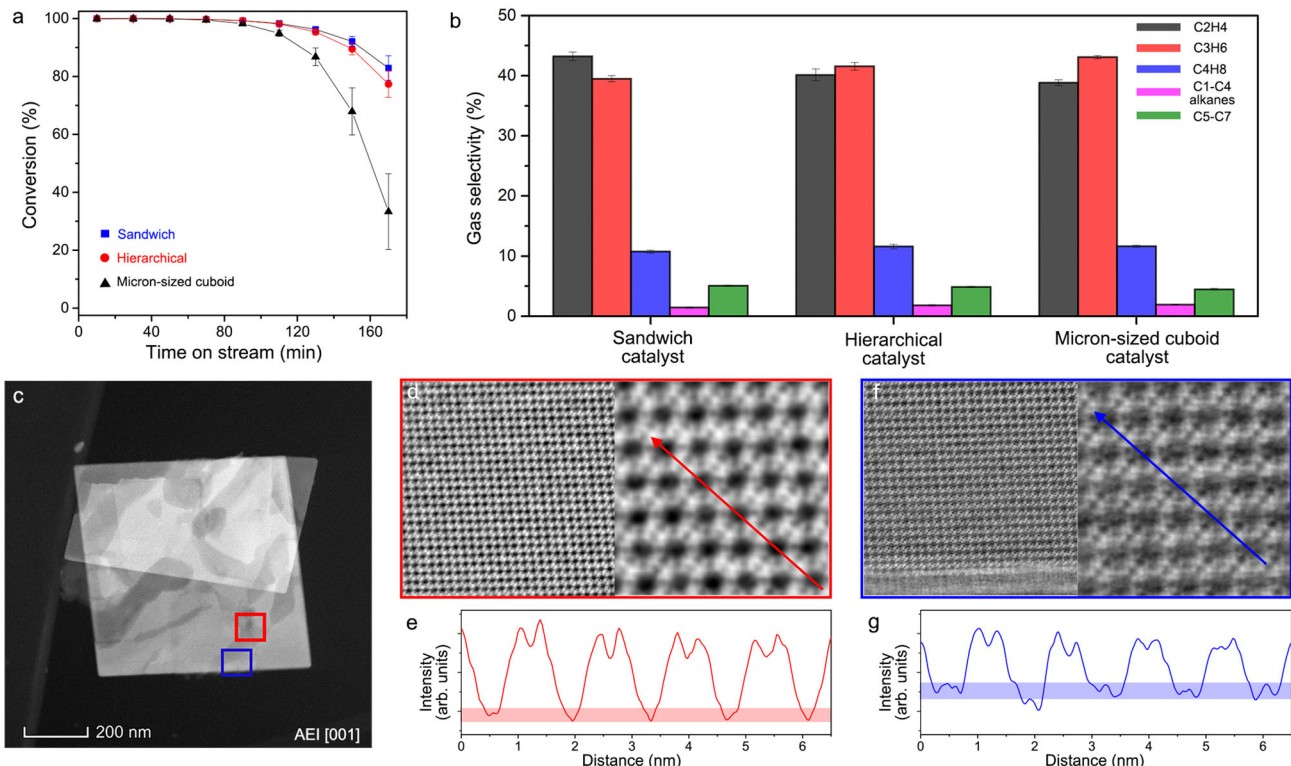

**Fig. 5 Catalytic performances of SAPO-34/18 catalysts. a** Conversion rates of methanol over three SAPO-34/18 catalysts. The blue, red, and black dots represent the results of sandwich, hierarchical, and micron-sized cuboid catalysts respectively. **b** Cumulative gas selectivity of the main products in the methanol conversion over three SAPO-34/18 catalysts, including light olefins, light alkanes, and heavy hydrocarbons. **c** HAADF-STEM image of a sandwich crystal viewed from the AEI [0 0 1] projection. **d–g** iDPC-STEM images and the corresponding intensity profiles showing the coking at the areas inside the bulk (**d**, **e**) and near the surface (**f**, **g**), respectively. The error bars in (**a**, **b**) represent the standard deviations of four sets of data in repeated experiments.

high conversion rate (>95%) for 130 min. These catalysts show very high gas selectivity of light olefins ($C_2$–$C_4$ olefins, ~93%), while very few alkanes ($C_1$–$C_4$ alkanes, ~2%) and heavy hydrocarbons ($C_5$–$C_7$, ~5%) were produced.

Based on a discrete Ising model[54], we have noticed that the preferential coking in the cages near surfaces will change the cage connectivity and make inner cages deactivated like the principles in the game of Go. Here, the imaging results provide the evidence for the coking behaviors. Figure 5c shows the HAADF-STEM image of a sandwich crystal after the deactivation. We used the iDPC-STEM and corresponding profile analysis to further resolve the areas marked by the red and blue frames in Fig. 5d–g. We can identify the contrasts of cokes in the cages near surfaces (Fig. 5f), while inner cages are nearly empty with few detectable species (Fig. 5d). The comparison of the intensity profiles in Fig. 5e, g also indicates that there are obvious peaks in the cages near surfaces instead of the inner ones. Therefore, a little coke in outer cages drastically reduced the activity of porous zeolite catalysts in spite of the inner empty cages that have not been deactivated, which is consistent with the Go-inspired Ising model.

As we have observed, the hierarchical pores and lamellas in the modified catalysts will provide additional external surfaces compared with the micron-sized cuboid catalyst. Thus, the modified catalysts could use more cages than the micron-sized cuboid catalyst until all the cages near external surfaces were filled by cokes. A longer dwell time of gas products in the micron-sized cuboid catalyst also accelerated carbon deposition in outer cages during their diffusion. Since the hierarchical catalysts still have solid areas with sizes of several hundred nanometers, it was deactivated a little faster than the sandwich catalyst. Therefore, this Ising model explains the size effect on the lifetime differences

between the catalysts with diverse architectures. Meanwhile, such diverse architectures can also explain the increase of propylene/ ethylene ratio in three catalysts, which results from the olefin cycle extended by the increasing dwell time of gas products with the sizes of inside solid areas (micron-sized cuboid > hierarchical > sandwich).

In summary, the iDPC-STEM is proved to be efficient for the imaging of the SAPO-34 and SAPO-18 frameworks with an atomic resolution. The Al, P, and O elements can be imaged with high contrasts, and the different projected lengths of bonds between them can be identified directly from the images. These observations help us to visually reveal the stacking and connection modes on the SAPO-34/18 interfaces. Then, two different SAPO-34/18 catalysts with the sandwich and hierarchical architectures are atomically imaged and studied by the iDPC-STEM. It is unraveled that their different architectures are mainly attributed to the different stacking sequences and distribution of SAPO-34 and SAPO-18 domains. By a series of syntheses, we find that increasing the content of Si elements is beneficial to the evolution from SAPO-18 to SAPO-34 lattices, and the TEAOH template can promote the increase of Si content in frameworks to direct the SAPO-34 lattices. These results help us to understand how different conditions affect the microstructures and local domains of zeolite crystals, which further leads to the sandwich and hierarchical SAPO-34/18 catalysts, respectively.

These catalysts are applied to the MTO catalysis and show high conversion of methanol and high selectivity of light olefins. Based on the direct imaging of hydrocarbon pools, we provide a probable deactivation mechanism of porous catalysts induced by surface coking, which will further inspire us how to design catalyst architectures for a longer lifetime. Thus, the controllable

content and distribution of Si elements (by changing the Si/Al ratios in feed and templates) could modify the stacking sequences in intergrowths and induce various macroarchitectures and corresponding catalytic performances of zeolite catalysts. Moreover, the iDPC-STEM technique in this work naturally can be applied to other more inorganic or organic/inorganic hybrid materials to provide the atomic information of inside light elements, such as the zeolites, perovskites, and MOFs, which may deeply promote their important applications in the catalysis, gas separation, and energy storage.

## Methods

**The synthesis of SAPO-34/18 intergrowth catalysts.** The SAPO-34/18 intergrowth catalysts were obtained by a hydrothermal method. First, the gel mixture, containing the $SiO_2$ (silica sol), $Al_2O_3$ (pseudoboehmite), $H_3PO_4$, and template, was prepared for further hydrothermal process. For the sandwich catalyst, we used the mixed dual templates of TEA and TEAOH, and the molar composition of the gel mixture is $Al_2O_3$:$0.17SiO_2$:$0.9P_2O_5$:$1.2TEA$:$0.35TEAOH$:$50H_2O$. This catalyst crystallized at 140 °C for 2 h, and then at 185 °C for another 8 h. For the hierarchical catalyst, we only used the single TEA template, and the molar composition of the gel mixture is $Al_2O_3$:$0.22SiO_2$:$1.06P_2O_5$:$3TEA$:$60H_2O$. This catalyst crystallized at 140 °C for 2 h, and then at 195 °C for another 6 h. For the classical micron-sized cuboid catalyst, we also used the single TEA template but a higher Si/Al ratio in feed, and the molar composition of the gel mixture is $Al_2O_3$:$0.64SiO_2$:$0.96P_2O_5$:$2TEA$:$53H_2O$. This catalyst crystallized at 140 °C for 2 h, and then at 185 °C for another 11 h. As we have mentioned in the main text, the cuboid catalyst can also be synthesized under low Si content by increasing crystallization time and temperature. During the catalytic tests, in order to keep the same Si/Al ratio in three catalysts, we used the micron-sized cuboid catalyst with a low Si/Al ratio. This catalyst was also synthesized by the TEA template and the molar composition of $Al_2O_3$:$0.25SiO_2$:$1.06P_2O_5$:$3TEA$:$60H_2O$. This catalyst directly crystallized at 195 °C for 10 h. After crystallization, the powder catalysts were obtained by centrifuging, and then washed several times with deionized water. Finally, the solid catalysts were dried at 100 °C for 12 h, followed by the calcination in air at 500 °C for 5 h to remove the templates.

To study the effect of templates on intergrowths, we use different combinations of TEA and TEAOH to direct SAPO-34/18 catalysts while other conditions remain unchanged (the Si/Al ratio in feed is the same as that of the sandwich sample). The molar composition of the gel mixture is $Al_2O_3$:$0.17SiO_2$:$0.9P_2O_5$:$xTEA$:$yTEAOH$:$50H_2O$. In this work, five combinations of TEA and TEAOH are $x1 = 0$, $y1 = 1.6$; $x2 = 0.35$, $y2 = 1.2$; $x3 = 0.8$, $y3 = 0.8$; $x4 = 1.2$, $y4 = 0.35$; $x5 = 1.6$, $y5 = 0$. These catalysts crystallized at 140 °C for 2 h, and then at 185 °C for another 8 h. Then, to study the effect of Si content on intergrowths, we only change the Si content in feed under the fixed conditions (including the contents of other reactants and dual templates). The molar composition of the gel mixture is $Al_2O_3$:$zSiO_2$:$0.9P_2O_5$:$1.2TEA$:$0.35TEAOH$:$50H_2O$. In this work, three Si contents in feed are $z1 = 0.1$; $z2 = 0.17$; $z3 = 0.3$. These catalysts also crystallized at 140 °C for 2 h, and then at 185 °C for another 8 h. After the same processes of filtration, washing, drying, and calcination, we can obtain the powder catalysts under different synthesis conditions.

**The iDPC-STEM imaging.** The iDPC-STEM imaging was conducted using a Cs-corrected (S)TEM (FEI Titan Cubed Themis G2 300). The operating voltage is 300 kV. The instrument is equipped with a DCOR+ spherical aberration corrector for the scanning probe that is aligned by a standard gold sample before imaging. The following aberration coefficients were used: $A1 = 1.47$ nm; $A2 = 6.07$ nm; $B2 = 5.4$ nm; $C3 = -91.2$ nm; $A3 = 225$ nm; $S3 = 66.6$ nm; $A4 = 1.57$ μm, $D4 = 2.24$ μm, $B4 = 2.3$ μm, $C5 = 615$ μm, $A5 = 170$ μm, $S5 = 34$ μm, and $R5 = 34$ μm, assuring a 60 pm resolution under a convergence semi-angle of 23.6 mrad. The convergence semi-angle for the iDPC-STEM is 15 mrad. The collection angle for the iDPC-STEM is 5–26 mrad. The beam current can be measured by the Faraday cup. The used beam current for the iDPC-STEM imaging is lower than 0.1 pA (measurement limit of beam current by the Faraday cup). Then, the corresponding dwell time of probe scanning is 32 μs/pixel with the pixel size of 0.1257 Å. Based on the beam current, dwell time, and pixel size, the calculated dose is $<1266$ e$^-$/Å$^2$.

**The MTO catalytic experiments.** The catalytic performances of different SAPO-34/18 catalysts were examined by MTO reaction. The experiments were carried out in a fixed-bed reactor (inner diameter of 8.00 mm). The catalysts (6.75 mg) mixed with quartz sand (40 mg) were activated under flowing $N_2$ at 500 °C for 2 h. Then, they were cooled to the reaction temperature of 450 °C at atmospheric pressure. The inlet gas was composed of 6% methanol and 94% $N_2$ in concentration. Here $N_2$ serves two functions as carrier gas and diluent gas. A feed of $N_2$ was saturated with gaseous methanol by flowing it through a methanol bottle that was placed in a thermostatic bath and then diluted by another feed of $N_2$ to keep WHSV = 4.0 h$^{-1}$. The product gas was injected and kept in several chromatography loops so that we can flexibly adjust the sampling interval to use by a gas chromatograph (Agilent 7890A) and a flame ionization detector to analyze the gas-phase product composition.

**Other characterizations.** The SEM images were captured by JSM-7401 high-resolution SEM (JEOL) with an operating voltage of 2–3 kV.

The HAADF-STEM imaging was conducted using the same aberration coefficients. The convergence semi-angle for the HAADF-STEM is 23.6 mrad. The collection angle for the HAADF-STEM is 51–200 mrad. The electron beam current for the HAADF-STEM is 50 pA. The dwell time for the HAADF-STEM is 16 μs/pixel.

The EDS mapping was also conducted using the same aberration coefficients. The beam current for the EDS is set between 10 and 15 pA to minimize the radiation damage to the specimens, and the dwell time is 4 μs/pixel with a map size of 256 × 256 pixels. A complete process of EDS mapping took roughly 0.5 h to reach enough SNR. The EDS data were integrated into the Velox software. Multi-polynomial and Brown–Powell models were employed to correct background and quantify the EDS data. The specimen tilt correction was applied automatically in the Velox software.

The PXRD results were obtained by a Rigaku D/Max-RB diffractometer using Cu Kα radiation at 40 kV and 20 mA.

The element contents (including Si/Al ratio) in the framework were obtained by an XRF spectrometer (SPECTRO XEPOS).

The Ar adsorption/desorption isotherms and corresponding Brunauer–Emmett–Teller surface areas of SAPO-34/18 catalysts were obtained by a surface area and porosity porosimetry (Micromeritics ASAP 2460).

The Hg intrusion porosimetry was performed using a Micromeritics AutoPore IV 9500 to investigate the distribution of mesopores and macropores in SAPO-34/18 catalysts. During the measurement, the operating pressure changed from 0.1 to 227.5 MPa. The pore size distribution was calculated by the Washburn equation based on the Hg volume injected in the pores under different pressures.

## Data availability

The authors declare that all relevant data supporting the findings of this study are available within the paper and its Supplementary information files. Additional data are available from the corresponding authors.

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

## Acknowledgements

This work was supported by the National Key Research and Development Program of China (2018YFB0604801) and the National Natural Science Foundation of China (No.20141301065, 21306103 and 22005170).

## Author contributions

B.S., X.C., Y.W., and F.W. conceived this project and designed the studies; B.S. and X.C. performed the electron microscopy experiments and data analysis; X.F. prepared the zeolite samples and performed the catalytic tests; H.X. and H.W. helped with the characterizations of zeolite samples. Y.W., F.W., and W.Q. helped with the data analysis. B.S. and X.C. wrote the paper with the contributions from all authors.

## Competing interests

The authors declare no competing interests.
