## [Peer Review File · Nature Communications]

Reviewers' Comments:

Reviewer #1:

Remarks to the Author:

Dear authors,

You provided a convincing insight into the performances of zeolite catalysts based on SAPO-34 and SAPO-18 frameworks. Altogether the impression of your manuscript, experimental work and conducted analysis is positive and encouraging.

Nevertheless, a few concerns and remarks remain and a few points unclear. Please address these several comments listed below before I can recommend your manuscript for publication:

1) You conducted an important in-depth analysis considering SAPO-34/18 intergrowth architectures for the MTO catalysis. Revealing specific distribution and sequences of SAPO-34 and SAPO-18 domains in the real space you show that the origin of the evolution of macro-architectures can be understood from the perspective of atoms and bonds. This seems to be very fundamental. Could you comment on how general this approach is? Maybe include some guidance and remarks on how it can be applied for other type of catalysts?

2) For your experimental analysis, you conducted a real space STEM imaging. For the overview of the structures you applied ADF-STEM (convergent semi-angle 23.6 mrad) and to reveal the fine structure of the atomic arrangement you selected direct phase imaging technique iDPC-STEM (convergent semi-angle 10 mrad). Could you clarify why did you use different convergent angles in these two types of images? Why using smaller convergence semi-angle (less resolution) for the fine structure imaging in iDPC-STEM (where higher resolution is wanted)?

3) In the "Methods" section you report 300kV high tension and corrector settings ensuring resolution of 60pm "under normal circumstances". Could you please specify which convergence semi-angle you consider "normal circumstances"? Although 60pm resolution is indeed ensured for the case for 23.6 mrad convergence semi-angle beam (theoretical maximal resolution 42pm), this is not the case for convergent semi-angle 10 mrad (theoretical maximal resolution 98pm). Therefore, please be specific.

4) A resolution inconsistency occurs in Fig. 2. where experimentally you report that "information transfer" goes to 94 pm, which is not possible to achieve with convergent semi-angle 10 mrad where maximal transfer possible is 98 pm. Your convergence semi-angle must have been more than 10 mrad, (most probably 12 to 15 mrad). Could it be that your convergence semi-angle measurement or calibration was inaccurate? Or a typo occurred? Could you please correct so that observations are consistent with the parameters?
This is related to the answers to questions 2) and 3).

5) You justified very well, by showing that Al, O and P atom columns can be distinguished that you can show the different projected lengths of Al(Si)-O and P(Si)-O bonds (when enriched with Si). This is indeed an excellent point! The one of the main advantages of iDPC-STEM is indeed ability to show all the elements across the periodic table and light (including hydrogen!) and heavy elements together. Please stress this further by including the following references that report these abilities:

[] E.G.T. Bosch, I. Lazić, S. Lazar, "Integrated Differential Phase Contrast (iDPC) STEM: A New Atomic Resolution STEM Technique To Image All Elements Across the Periodic Table", *Microsc. Microanal.* 22 (Suppl 3), 2016
doi:10.1017/S1431927616002385

[] D. Song, X. Zhang, C. Lian et.al. "Visualization of Dopant Oxygen Atoms in a Bi₂Sr₂CaCu₂O_{8+δ}", *Advanced Functional Materials* 2019, 1903843

doi: 10.1002/adfm.201903843

[] S. de Graaf, J. Momand, C. Mitterbauer, S. Lazar, B.J. Kooi, "Resolving hydrogen atoms at metal-metal hydride interfaces", *Science advances* 6 (5), (2020), eaay4312

6) Important point in your analysis is imaging of beam sensitive structures, such as Zeolites. Please include the following references on first Zeolites imaging and recent work on MOFs.

[] Lazić, I., Bosch, E. G. T., Lazar, S., Wirix, M. & Yücelen, E. "Integrated differential phase contrast (iDPC)-direct phase imaging in STEM for thin samples" *Microsc. Microanal.* 22(Suppl_3), pp36–pp37 (2016).

[] B. Shen, X. Chen, K. Shen, H. Xiong & F. Wei "Imaging the node-linker coordination in the bulk and local structures of metal-organic frameworks" (2020) 11:2692
<https://doi.org/10.1038/s41467-020-16531-y>

7) One of the important beam parameters as function of convergent semi-angle is depth of focus of the probe. It must be related to the sample thickness along the imaging direction to ensure proper interpretation. Please see and include the relevant reference:

[] E.G.T Bosch, I. Lazić, "Analysis of depth-sectioning STEM for thick samples and 3D imaging", *Ultramicroscopy* 207, (2019) 112831

Please comment on the sample thickness of the critical regions where atomic resolution imaging takes place (especially Fig. 2). The depth of focus as function of opening angle is given by $2 \cdot \text{wavelength} / (\text{conv semi-angle})^2$ which for 10 mrad beam at 300kV (wavelength 2pm) gives 40nm which is good enough for the full projection imaging. This is probably the reason why smaller convergence angle was used for these regions (see also question 2)?

8) Although beam current and dwell time were specified in the "Methods", an actual dose applied in $e/\text{Å}^2$ should be computed and specified with the specific measurements: $(\text{beam current}) \cdot (\text{dwell time}) / (\text{pixel size})^2$.

9) In Fig 3b you show and refer to "uniformly mixed SAPO-34/SAPO-18". Nevertheless, on the single raw level, the switching between blue and red arrows and direction is not precisely uniform... It goes more like 1up, 2 down, then 1 up, 1 down... 1 up 3 down ... etc. Indeed, on the larger scale this would appear homogeneous but not strictly speaking uniform ... Just a remark.

Best regards

Reviewer #2:

Remarks to the Author:

In this manuscript the authors examine intergrowths between two silicoaluminophosphates (SAPOs) that are used as commercial catalysts for methanol to olefin (MTO) reactions: SAPO-34 with the CHA structure and SAPO-18 with the AEI structure. The overarching topic of this study is the analysis of intergrowths, which is a subject of interest in zeolite research, but also extending to a broader range of crystalline materials. The origins of intergrowths and their impact on applications (such as catalysis) are not well understood; therefore, advancements in this area of research hold promise for the ability to control intergrowths.

The use of integrated differential phase contrast scanning transmission electron microscopy (iDPC-STEM) in this manuscript results in absolutely stunning images with near atomic level resolution. Images such as those shown in Figure 2 provide some of the best examples of zeolite intergrowths. The authors have captured interesting phenomena in a range of samples that were prepared using two different organic structure-directing agents: triethylamine (TEA) and tetraethylammonium hydroxide (TEAOH). The opinion of this manuscript is twofold. The TEM work is truly beautiful and captures aspects of intergrowths that have not been previously reported. The authors are to be commended on achieving such superb resolution that show trends among different materials, thus providing insight into the formation of intergrowths. On the other hand, the roles of TEA and TEOH in directing intergrowths are not definitively proven, the impact of Si doping is ambiguous, and the catalysis experiments lack clear conclusions (i.e. in relation to structure-performance relationships). These three major concerns need to be addressed before this article should be considered for publication. Details of these comments, along with several minor comments, are provided below:

- On page 3 the authors make the following statement: “two kinds of SAPO-34/18 intergrowth catalysts were successfully synthesized by changing the templates and the contents of reactants.” It is not clear how TEA and TEOH direct different intergrowths. How many total experiments were performed? How reproducible are the syntheses? It wasn’t clear if the authors used pure TEOH, and in the case of mixtures of TEA and TEOH how many different combinations were tested? Did the authors see a progressive shift in intergrowths from pure TEA to pure TEOH?
- Mechanistic arguments made in Figure 3 are unclear. The random orientations shown in Fig. 3b appear similar to those in Fig. 3c and d, which seems to indicate that mesopore formation cannot be linked to the frequency of lattice mismatching. The authors need to provide more definitive evidence of characteristic features in the crystal structure that may lead to mesopore formation. Generalizing these trends based on limited examples is also problematic.
- The reviewer strongly disagrees with the statement on page 8: “the Si content of the SAPO-34 domain is significantly higher than that of the SAPO-18 domain, since the Si/Al ratio rises from ~ 0.05 to ~ 0.2 through the SAPO-18 to SAPO-34 interface.” The crystal used for EDS analysis has an odd shape. The center of the crystal appears to be much lower (i.e. thinner) than the two ends. This likely gives rise to the drop in elemental intensity in the center region. All elements have lower intensity, and it is likely the resolution for Si (the minor component) under these conditions is subject to error, giving rise to the lower Si/Al ration in Fig. 4b. Do the authors have examples of crystals that are level? The crystal in Fig. 3e is odd and seems to be an abnormality (unless the authors have shown this is representative of many crystals prepared in the synthesis batch. Also, what are the Si/Al ratios of pure SAPO-34 and SAPO-18 (these materials are not mentioned)? Does one traditionally have a different Si/Al ratio? This mechanistic argument requires more evidence – particularly if the authors are going to make the following definitive statement in the conclusion section: “the enrichment of Si element is directly related to the switch between the SAPO-34 and SAPO-18 lattices.”
- There are several comments and concerns with the catalysis section:
 - o There are no error bars in Figure 5. It is uncertain how the authors can claim differences in selectivity when the data differ by only $\pm 2\%$.
 - o The reviewer strongly disagrees with the statement on page 10: “the hierarchical catalysts lose their activity faster than the sandwich catalysts.” The data in Fig. 5c are nearly identical and it is difficult to see how there is a difference outside of experimental error (again no error bars are provided).
 - o The differences among catalysts with intergrowths are essentially nonexistent. As such, it is not clear what is learned from the catalytic studies. The statement in the conclusion, “MTO catalysis showing very high conversion rate and high selectivity of light olefins,” is commonly known and is not reflective of any unique features of the materials prepared in this study. Why did the authors not compare samples with intergrowths to pure SAPO-34 and SAPO-18? In the end, the only new information presented in this section is the coking analysis, which is quite interesting. To this end, the authors need to justify the need for including the MTO results.

Minor Comments:

- On page 5 the authors refer twice to "Fig. 3g and h", but it seems this should be in reference to Figure 2.
- In Fig. 2g and h, how do the authors know that atoms labelled Al and/or P cannot also be Si?
- In the caption for Fig. 4c, it is stated "crystal in Fig. 3" but there are two crystals shown in that figure. The authors should be more specific.
- In the caption of Fig. 5, the line starting with "e-g" should be "e-h".

Reviewer #3:

Remarks to the Author:

This manuscript describes how by integrated differential phase contrast (iDPC) STEM it is possible to study SAPO-34 and SAPO-18 intergrowths with atomic resolution and to relate changes in the synthesis conditions to the formation of different nanostructures. It provides detailed information about the stacking of these two zeolites and how the distribution of domains determines the development of different materials. This manuscript extends the use of iDPC STEM, recently used by the authors to image local structures in ZSM-5 DOI: 10.1002/adma.201906103, to the study of zeolitic intergrowths.

Thanks to their microscopic analysis, the authors have been able to conclude that the substitution of Si atoms induces the switch between the two zeolitic structures as they intergrow. It is especially interesting the work done on the structural analysis done in the AA/AB stacking selectivity and how the different elements (Si, Al and P) are distributed in the various intergrowths. Thanks to this atomic spatial analysis was possible to conclude that the Si distribution in the zeolite framework determines the mixing and separation between the SAPO-34 and SAPO-18 domains.

The authors describe so-called hierarchical structures including nano-cuboids and sandwich materials with nano-lamellas. These nanoarchitectures are studied in detail at atomic scale by advanced microscopy techniques, but poorly characterized at the mesoscale. For example, and despite that the authors talk about the presence of mesoporosity in the crystals, there are no gas adsorption isotherms, or any other analysis typically used to characterize porosity. From the micrographs (Figure 1a and Figure 3a) it is very difficult to conclude that there are mesopores (with diameters in the 2-50 nm range) in the zeolites. They look more like irregular voids inside the crystal, which will not qualify them as hierarchical, in the strict meaning of the term. The manuscript lacks a proper characterization of the different materials obtained beyond the excellent microscopy analysis done at atomic scale. The materials were tested for the catalytic conversion of methanol, as previously reported by the authors in a previous paper DOI: 10.1016/j.cattod.2014.03.038, and both their conversion and selectivities compared in bases on the shortened diffusion path-lengths present in the different materials prepared. The direction in the results support the hypothesis of the authors, for example the reduction of hydrogen transfer reactions found in the nano-lamellas and the higher conversion of ethylenes and propylenes into butylenes. However, the improvements are small compared to other studies using hierarchical zeolites.

Despite the exceptional quality and interest of the microscopy work done by the authors, I must recommend the rejection of this manuscript for publication in Nature Communications, and to suggest a more specialized journal, for the following reasons:

1. This study will be of interest to a narrow audience, which shouldn't be the case of an article of a journal that publishes results from a very broad range of topics.
2. The relationship found by the authors between the synthesis conditions and the formation of different intergrowths is interesting, but it is not evident that it will lead to the discovery of new significantly better catalysts.
3. The improvements in the catalytic conversion of methanol described in the final part of the manuscript are minor and do not constitute a significant breakthrough; although, they support the claims of the authors.
4. The different materials prepared by modifying the synthesis conditions lack a proper characterization, beyond the excellent microscopy work described in the manuscript.

Referee #1

Thank you for your recommendation of our manuscript. We are excited to see your very positive comments on the novelty and significance of this work. Your comments and suggestions really help us to improve our manuscript on the (S)TEM imaging. Based on your comments, we have made a revision on the manuscript and answered all your questions as follows. After this revision, we sincerely ask you to reconsider our revised manuscript for the further publication.

1) You conducted an important in-depth analysis considering SAPO-34/18 intergrowth architectures for the MTO catalysis. Revealing specific distribution and sequences of SAPO-34 and SAPO-18 domains in the real space you show that the origin of the evolution of macro-architectures can be understood from the perspective of atoms and bonds. This seems to be very fundamental. Could you comment on how general this approach is? Maybe include some guidance and remarks on how it can be applied for other type of catalysts?

Reply: Thank you for your comment on this point. The iDPC-STEM has been proved to be an efficient tool for the atomic characterizations of beam sensitive materials and light elements with a high resolution and SNR. Many catalysts used in heterogeneous catalysis are sensitive to electron beam and consist of light elements, such as various zeolites, 2D layered materials and MOFs. In previous works, framework structures of ZSM-5, MIL-101 and UiO-66 have been atomically imaged and analyzed by the iDPC-STEM. Therefore, this general approach to resolve and understand catalyst structures can be applied into various catalyst systems. Although the XRD and other spectroscopic methods have contributed to the structural analysis of catalysts, some local structures, such as intergrowth and twin interface, should be better studied by real-space imaging methods. These local structures are very common in heterogeneous catalysts, which are directly related to the macro-architectures and catalytic performances of these crystals. For porous materials, the iDPC-STEM imaging results help us to reveal the geometry and topology of local channel systems, which deeply affect the adsorption, transport and reaction of guest molecules in catalysis. We have added related comments in the “Conclusions” section of the revised manuscript.

2) For your experimental analysis, you conducted a real space STEM imaging. For the overview of the structures you applied ADF-STEM (convergent semi-angle 23.6 mrad) and to reveal the fine structure of the atomic arrangement you selected direct phase imaging technique iDPC-STEM (convergent semi-angle 10 mrad). Could you clarify why did you use different convergent angles in these two types of images? Why using smaller convergence semi-angle (less resolution) for the fine structure imaging in iDPC-STEM (where higher resolution is wanted)?

Reply: Thank you for your questions. First of all, as you also pointed out in question 4), the convergent semi-angle of iDPC-STEM is inaccuracy (a typo). We have checked the imaging parameters again, and found that the convergent semi-angle of iDPC-STEM is 15 mrad. These two convergent semi-angles for ADF- and iDPC-STEM are proved to be the suitable parameters in our practice to achieve best

image quality in two different imaging modes. In the iDPC-STEM, a smaller convergent semi-angle of 15 mrad will bring deeper information as you also mentioned in question 7). The diffraction limit with a convergent semi-angle of 15 mrad is about 66 pm, which is enough for us to obtain atomic information of zeolite frameworks.

3) In the “Methods” section you report 300kV high tension and corrector settings ensuring resolution of 60pm “under normal circumstances”. Could you please specify which convergence semi-angle you consider “normal circumstances”? Although 60pm resolution is indeed ensured for the case for 23.6 mrad convergence semi-angle beam (theoretical maximal resolution 42pm), this is not the case for convergent semi-angle 10 mrad (theoretical maximal resolution 98pm). Therefore, please be specific.

Reply: Thank you for your suggestion. The highest resolution (60 pm) was obtained under the convergence semi-angle of 23.6 mrad (ADF-STEM). We have specified it in the “Methods”.

4) A resolution inconsistency occurs in Fig. 2. where experimentally you report that “information transfer” goes to 94 pm, which is not possible to achieve with convergent semi-angle 10 mrad where maximal transfer possible is 98 pm. Your convergence semi-angle must have been more than 10 mrad, (most probably 12 to 15 mrad). Could it be that your convergence semi-angle measurement or calibration was inaccurate? Or a typo occurred? Could you please correct so that observations are consistent with the parameters? This is related to the answers to questions 2) and 3).

Reply: Thank you for your questions. We have checked the imaging parameters again, and found that the convergent semi-angle of iDPC-STEM is 15 mrad, not 10 mrad. We are very sorry about this typo in the manuscript. To confirm the convergence semi-angle, we give the screen shot of Velox software to show the images and corresponding values of imaging parameters.

We noticed that the convergent semi-angle (beam convergence) should be 15 mrad. We also updated the other parameters based on the information of this image.

5) You justified very well, by showing that Al, O and P atom columns can be

distinguished that you can show the different projected lengths of Al(Si)-O and P(Si)-O bonds (when enriched with Si). This is indeed an excellent point! The one of the main advantages of iDPC-STEM is indeed ability to show all the elements across the periodic table and light (including hydrogen!) and heavy elements together. Please stress this further by including the following references that report these abilities:

Reply: Thank you for your very positive comments on this point. We have added the related publications you mentioned in the reference [35,39,40].

6) Important point in your analysis is imaging of beam sensitive structures, such as Zeolites. Please include the following references on first Zeolites imaging and recent work on MOFs.

Reply: Thank you for your comments. We have added the related publications you mentioned in the reference [36,44].

7) One of the important beam parameters as function of convergent semi-angle is depth of focus of the probe. It must be related to the sample thickness along the imaging direction to ensure proper interpretation. Please see and include the relevant reference: Please comment on the sample thickness of the critical regions where atomic resolution imaging takes place (especially Fig. 2). The depth of focus as function of opening angle is given by $2 \cdot \text{wavelength} / (\text{conv semi-angle})^2$ which for 10 mrad beam at 300kV (wavelength 2pm) gives 40nm which is good enough for the full projection imaging. This is probably the reason why smaller convergence angle was used for these regions (see also question 2)?

Reply: Thank you for your comments. We have added the related publications you mentioned in the reference [38]. Then, the convergent semi-angle of 15 mrad (corrected) is corresponding to the depth of focus of about 18 nm. The SAPO-18 areas in sandwich crystals are not flat on the surface (irregular shape) and the imaging area in Fig. 2 is just at the edge of SAPO-18 area. Thus, it can be guaranteed that the thickness is smaller than 10 nm, and the depth of focus of about 18 nm is good enough for the full projection imaging.

8) Although beam current and dwell time were specified in the “Methods”, an actual dose applied in $e^-/\text{\AA}^2$ should be computed and specified with the specific measurements: $(\text{beam current}) \cdot (\text{dwell time}) / (\text{pixel size})^2$.

Reply: Thank you for your suggestion. Based on the beam current, dwell time and pixel size, the calculated dose is $< 1266 e^-/\text{\AA}^2$. Here, the $<$ comes from the measurement limit of beam current by the Faraday cup. Thus, we believe the actual dose is much lower than this value. However, for the inorganic zeolites, the critical electron dose for the structural collapse (thousands of $e^-/\text{\AA}^2$) is much higher than that of MOFs (about $50 e^-/\text{\AA}^2$). Thus, this electron dose is enough to protect the zeolite structures from the obvious damages for us to resolve the atomic structures.

9) In Fig 3b you show and refer to “uniformly mixed SAPO-34/SAPO-18”. Nevertheless, on the single raw level, the switching between blue and red arrows and

direction is not precisely uniform... It goes more like 1 up, 2 down, then 1 up, 1 down... 1 up 3 down ... etc. Indeed, on the larger scale this would appear homogeneous but not strictly speaking uniform ... Just a remark.

Reply: Thank you for your suggestion. Your comment is right that the arrangement of SAPO-18 and SAPO-34 is not precisely uniform, and it can be expressed as “highly mixed SAPO-34/SAPO-18”. We have changed the expression about it in Fig. 3 in the revised manuscript.

Referee #2

Thank you for your recommendation of our manuscript. We are excited to see your very positive comments on the novelty and significance of this work. Your comments and suggestions show your high-degree expertise on zeolite catalysts and really help us to improve our manuscript efficiently. Based on your comments and suggestions, we made a revision on the manuscript and answered all your questions and suggestions as follows. After this revision, we sincerely ask you to reconsider our revised manuscript for the further publication.

1) On page 3 the authors make the following statement: “two kinds of SAPO-34/18 intergrowth catalysts were successfully synthesized by changing the templates and the contents of reactants.” It is not clear how TEA and TEOH direct different intergrowths. How many total experiments were performed? How reproducible are the syntheses? It wasn't clear if the authors used pure TEOH, and in the case of mixtures of TEA and TEOH how many different combinations were tested? Did the authors see a progressive shift in intergrowths from pure TEA to pure TEOH?

Reply: Thank you for your questions. To confirm the effect of TEOH/TEA dual templates on zeolite synthesis, we use different combinations of TEA and TEOH to direct SAPO-34/18 intergrowth while other conditions remain unchanged (the Si/Al ratio in feed is the same as that of the sandwich sample). Five different combinations are tested including pure TEOH, pure TEA and mixtures with TEA/TEOH ratios of 0.29, 1 and 3.43. The PXRD results of these samples (Fig. R1) provide a valid evidence of the change in SAPO-34/18 ratios. As shown in Fig. R1, we can identify SAPO-18 domain according to the characteristic AEI peaks at 2theta of 10.7° and 17.0° (corresponding to the AEI (111), (022) and (113) planes). At high TEOH content (TEA/TEOH ratios of 0 and 0.29), we can hardly observe these two AEI peaks in the PXRD curves. The SAPO-34 domains are dominant, while the SAPO-18 lattices only exist as stacking faults in these SAPO-34 domains. However, with TEA increased, these two peaks can be observed and the increase of peak heights indicate that there are more SAPO-18 domains in these intergrowths. Thus, we have observed a progressive shift of two domains from pure TEA to pure TEOH. These results show that at such Si content, pure TEOH is usually used to direct zeolite structures to SAPO-34, and pure TEA will induce SAPO-18 domain to form intergrowths. Such effect of TEOH and TEA dual templates was also discussed preliminarily in previous works (Wang Y, Chen S L, Jiang Y J, et al. *RSC Adv.* 2016, 6:104985-104994; Wang P, Lv A, Hu J, et al. *Micropor Mesopor Mater.* 2012, 152:178-184).

Figure R1. PXRD results of SAPO-34/18 intergrowth directed by different combinations of TEAOH and TEA dual templates.

Meanwhile, we also use the iDPC-STEM imaging as a complementary method to the PXRD to confirm such structure differences by changing template combinations. The observations of channels and pores can also help us to identify the different composition of SAPO-34/18 in the samples directed by pure TEA, pure TEAOH and TEA/TEAOH mixtures respectively. As shown in Fig. R2, the sample directed by pure TEAOH is dominant by the SAPO-34 domain (AA stacking) with a few SAPO-18 (AB stacking) faults, while the sample directed by pure TEA is just the opposite (more SAPO-18). As for the sample directed by TEA/TEAOH mixture (TEA/TEAOH ratio of 3.43 for example), we can observe the mixture of SAPO-34 domains and SAPO-18 domains inside. These imaging results are consistent with the PXRD results, which confirms the effect of TEAOH and TEA dual templates as we mentioned above. Combining with the effect of Si content in feed (in the reply to question #3), we can understand how the sandwich samples are formed. In the synthesis of sandwich catalysts (at given Si content), the mixed TEA and TEAOH templates preferentially induced the synthesis of SAPO-34 and SAPO-18 domains respectively, where the SAPO-34 domains form the lamellas linked by the SAPO-18 domains.

Thanks to your concerns and suggestions, we also realized that it is necessary to confirm such law by a variable-controlling method for a better understanding on the structural modification of SAPO-34/18 intergrowth. We have added two figures and related discussions in Fig. 3 and Fig. S9.

Figure R2. IDPC-STEM images of samples directed by pure TEAOH, TEAOH/TEA mixture (TEA:TEAOH=3.43) and pure TEA respectively. The corresponding PXRD results are given in Fig. R1. At this Si content in feed, the TEAOH-template sample is dominant by SAPO-34, while the TEA-template sample is dominant by SAPO-18. The sample directed by dual TEAOH/TEA template is a mixture of SAPO-34 and SAPO-18 domains.

- 2) Mechanistic arguments made in Figure 3 are unclear. The random orientations shown in Fig. 3b appear similar to those in Fig. 3c and d, which seems to indicate that mesopore formation cannot be linked to the frequency of lattice mismatching. The authors need to provide more definitive evidence of characteristic features in the crystal structure that may lead to mesopore formation. Generalizing these trends based on limited examples is also problematic.

Reply: Thank you for your comments. There is a hierarchical pore system, including mesopores, macropores and hollow areas, in the hierarchical catalysts. These pores can be confirmed by the STEM imaging and Hg intrusion in Fig. S1 and S2. These mesopores can be roughly divided into small mesopores (<10 nm) and large ones (>10nm), which may be caused by different reasons. For the large ones (>10 nm), they cannot be simply explained by the lattice mismatching. It is more likely that different growth rates between SAPO-34 and SAPO-18 along $\beta\gamma$ plane will slow down the overall crystallization of hierarchical catalysts so that the separated parts cannot link in a limited time and just form large mesopores, macropores and even larger hollow areas.

For the small mesopores (<10 nm), we studied the lattices near small pores which can be recorded in one image in Fig. 3f and g. We have observed many evidences of lattices around the small mesopores. As you suggested, we added more evidential images in Fig. S8 (as shown in Fig. R3) to confirm this characteristic structure for these mesopores. Then, the discussions on the formation of small

mesopores is given based on the following facts:

1. Such mesopores, macropores and hollow areas are only formed in the samples synthesized with low Si content and pure TEA template. They will not exist in those synthesized with a high Si content (form solid particles) and also in the sandwich samples.
2. Based on the PXRD and imaging results, we can resolve the local structures and stacking sequences of these modified catalysts. SAPO-34 and SAPO-18 lattices are highly mixed in the hierarchical catalysts without definable domains and interfaces. While, SAPO-34 and SAPO-18 domains are completely separated in the sandwich catalysts.
3. According to the images in Fig. R4, all mesopores are accompanied by the high-frequency mismatch between the upper and lower lattices around these pores.
4. There is no mesoporous structure in all the samples with separate or large-size domains, such as the sandwich and cuboid catalysts.

According to these facts, we find that the formation of small mesopores is closely related to the lattice mismatching. The growth of new layers in large crystals may start from different sites of surface layers, and these separated fragments gradually connect to form a complete layer. The frequent switches between the AA and AB stacking make the lattice mismatch in the same layer more likely to exist than that in single large-size domain, and then cracks are generated to form small mesopores (<10 nm) in Fig. R3. This is easy to understand. For example, if the probabilities of AA and AB stacking are the same during the crystal growth, the separated fragments of new layer on the surface will have a half probability of being mismatched (AA vs AB). However, in a single domain (whether it is SAPO-34 and SAPO-18), the separated fragments of new layer on the surface will maintain the same stacking mode with the surface layer, thus, without the in-layer mismatch.

Meanwhile, the image you mentioned (Fig. 3e in revised manuscript) is captured from a thin area near a large mesopore (or macropore or hollow area) as we marked in Fig. 3d. We can give the complete image of this figure in Fig. R4. We clip this image to draw the arrows just above the image to indicate the stacking sequence of SAPO-34 and SAPO-18. Therefore, we can observe a large pore near this area, and the reason why there is no pore in this solid area is that these lattices were generated from one site on each layer. That is also why there is always solid area in the center of hierarchical catalysts. We have revised the related discussions in the section:” Resolving the micro-structures in different SAPO-34/18 catalysts”.

Figure R3. More iDPC-STEM images showing the mismatched lattices surrounding the through mesopores in SAPO-34/18 intergrowths. These images confirm the correlation between the mesopore formation and lattice mismatching.

Figure R4. Complete raw iDPC-STEM image of Fig. 3e.

- 3) The reviewer strongly disagrees with the statement on page 8: “the Si content of the SAPO-34 domain is significantly higher than that of the SAPO-18 domain, since the Si/Al ratio rises from ~ 0.05 to ~ 0.2 through the SAPO-18 to SAPO-34 interface.” The crystal used for EDS analysis has an odd shape. The center of the crystal appears to be much lower (i.e. thinner) than the two ends. This likely gives rise to the drop in elemental intensity in the center region. All elements have lower intensity, and it is likely the resolution for Si (the minor component) under these conditions is subject to error, giving rise to the lower Si/Al ratio in Fig. 4b. Do the authors have examples of crystals that are level? The crystal in Fig. 3e is odd and seems to be an abnormality (unless the authors have shown this is representative of many crystals prepared in the synthesis batch. Also, what are the Si/Al ratios of pure SAPO-34 and SAPO-18 (these materials are not mentioned)? Does one traditionally have a different Si/Al ratio? This mechanistic argument requires more evidence – particularly if the authors are going to make the following definitive statement in the conclusion section: “the enrichment of Si

element is directly related to the switch between the SAPO-34 and SAPO-18 lattices.”

Reply: Thank you for your questions. Since the difference of thickness in sandwich sample is inevitable (this is the characteristic architecture of sandwich sample), the intensities of all elements indeed decrease in the center area. Such crystal in the sandwich sample is not abnormal. The representativeness and repeatability has been shown in more STEM images of the sandwich crystals prepared in the same batch in Fig. S3 (as shown in Fig. R5).

Figure R5. STEM images of the sandwich SAPO-34/18 catalysts. The thickness of lamellas can be measured as 10-100 nm.

However, we believe that the noises in EDS mapping will not affect our conclusions. On the one hand, we have done the elemental analysis not only in profiles but also in areas given in Fig. S10 (as shown in Fig. R6 and Table R1). We give the atomic fractions and errors of different elements in three marked areas in Table R1 by EDS mapping. In the center area, the average Si atomic fraction is 0.62 ± 0.16 , while those in the both sides are 2.75 ± 0.60 and 2.66 ± 0.58 . Therefore, the Si signal is not subject to error, and the differences of Si content (also the Si/Al ratio) in these three areas are significant enough to come to our conclusion about the Si distribution. On the other hand, in the dash-line profile in Fig. 4d (indicating the Si/Al ratio), the change of Si/Al ratio is obvious enough and there is a four-fold difference between the Si/Al ratios in center area and both sides (0.05 and 0.20 respectively).

Figure R6. EDS mapping of the sandwich SAPO-34/18 crystal studied in Fig. 3 and 4. The proportions of different elements in three areas are given in Table R1.

Table R1. The proportions of different elements in three areas in Fig. R6.

	Element	Atomic Fraction (%)	Atomic Error (%)	Mass Fraction (%)	Mass Error (%)
Area1	O	66.2	6.42	52.22	3.21
	Al	17.49	3.85	23.26	4.82
	Si	2.75	0.6	3.81	0.78
	P	13.56	2.89	20.71	4.14
Area2	O	65.56	6.84	51.36	3.39
	Al	17.95	4.01	23.71	4.94
	Si	0.62	0.16	0.86	0.2
	P	15.87	3.44	24.07	4.85
Area3	O	65.91	6.43	51.88	3.2
	Al	17.55	3.87	23.3	4.82
	Si	2.66	0.58	3.68	0.75

	P	13.88	2.97	21.15	4.23
--	---	-------	------	-------	------

Furthermore, we added new experiments to investigate the effect of Si content on the SAPO-34/18 selectivity. In a series of syntheses, we only change the Si content in feed under the fixed conditions (including the contents of other reactants and dual templates). The results are given in Fig. 4a (as shown in Fig. R7). We can observe a selectivity of SAPO-34 and SAPO-18 depending on different Si contents in feed. According to the changes of characteristic AEI peaks at 2theta of 10.7° and 17.0° , it can be identified that, with the increase of Si/Al ratio, the content of SAPO-18 lattices decreases gradually, and there seems to be only SAPO-34 at the Si/Al ratio of 0.15 in feed. This is a solid evidence for our point of view that higher Si content in the initial gel mixture will preferentially induce the formation of SAPO-34, while lower Si content is favorable for SAPO-18. Such results are consistent with previous works on the synthesis of SAPO-34/18 intergrowths with different Si contents (Sun Q, Ma Y, Wang N, et al. *J Mater Chem A*, 2014, 2:17828-17839; Smith R L, Svelle S, Campo del P, et al. *Appl Catal A-Gen*, 2015, 505:1-7.).

Figure R7. PXRD results of SAPO-34/18 intergrowth synthesized with different Si contents in feed.

Meanwhile, as we mentioned above, we investigate the selectivity of SAPO-34/18 using different combinations of TEA/TEAOH templates. The samples directed by pure TEAOH and TEA are mainly composed of SAPO-34 and SAPO-18 domains respectively. Although the Si content in feed is the same, the real Si/Al ratio in the TEAOH-template framework (0.16, measured by the X-ray fluorescence analysis)

is higher than that in the TEA-template framework (0.11). This result may answer your question if SAPO-34 and SAPO-18 usually have different Si/Al ratios. A more precise expression is that, under the same feeding conditions, the obtained SAPO-34 sample will have a higher Si/Al ratio in framework than the SAPO-18 sample. These results together reveal the influence of Si element on the structural selectivity in SAPO-34/18 intergrowth. It can be interpreted as that cationic groups in TEAOH show a stronger affinity to the negative Si-O-Al units in inorganic segments in the gel mixture, and more Si acid sites are integrated into the bulk frameworks to form SAPO-34 domains preferentially. Thanks to your comments and suggestions, we added these new experiments and discussions in the revised manuscript. We believe that these supplements will provide us a new insight into the mechanism about the controllable synthesis of SAPO-34/18 intergrowth.

- 4) There are no error bars in Figure 5. It is uncertain how the authors can claim differences in selectivity when the data differ by only $\pm 2\%$.
- 5) The reviewer strongly disagrees with the statement on page 10: “the hierarchical catalysts lose their activity faster than the sandwich catalysts.” The data in Fig. 5c are nearly identical and it is difficult to see how there is a difference outside of experimental error (again no error bars are provided).

Reply: Thank you for your questions. These two questions (#4 #5) can be addressed together. We have examined the catalytic performances of SAPO-34/18 catalysts again. Four sets of repeated experiments are carried out on each sample under the same conditions to estimate the error and repeatability. The results (including the curves of micron-sized cuboid sample) are given in Fig. R8. Based on the data and error bars, we clearly identify the differences of the catalytic performances between three catalysts outside of experimental errors. For the conversion rate, the micron-sized cuboid sample is obviously deactivated faster than the modified catalysts (the hierarchical and sandwich samples). And the sandwich sample will maintain a little longer lifetime than the hierarchical sample that is outside of experimental errors. Meanwhile, we also observe different propylene/ethylene ratios in the gas products of three catalysts based on the data of cumulative gas selectivity. These results can be explained consistently by different sizes of solid areas in three catalysts (micron-sized cuboid>hierarchical>sandwich). Please see details in the discussions on Fig. 5 in the revised manuscript (the section: “Catalytic performances of SAPO-34/18 catalysts”).

These comparisons indicate that reducing diffusion distances of gas products by structure modification is an efficient strategy to extend the lifetime of catalyst. After adding error bars, the catalytic results can satisfy our requirements for repeatability and strongly support our characterizations and conclusions. We sincerely thank you for these constructive comments, and this point is indeed a significant problem that we didn't take into full account. We have revised the Fig.

5, Fig. S13 and the related discussions on the catalytic performances of these catalysts.

Figure R8. Catalytic performances of SAPO-34/18 catalysts. **a**, Conversion of methanol over three SAPO-34/18 catalysts at different times on stream. The error bars represent the standard deviations of data in repeated experiments. **b**, Cumulative gas selectivity of the main products in the methanol conversion over three SAPO-34/18 catalysts, including the light olefins, light alkanes and heavy hydrocarbons. **c-h**, Gas selectivity of the main products during the methanol conversion. These catalysts show very high gas selectivity of light olefins (C_2 - C_4 olefins, ~93%), while a few alkanes (C_1 - C_4 alkanes, ~2%) and heavy hydrocarbons (C_5 - C_7 , ~5%) were produced.

- 6) The differences among catalysts with intergrowths are essentially nonexistent. As such, it is not clear what is learned from the catalytic studies. The statement in the conclusion, “MTO catalysis showing very high conversion rate and high selectivity of light olefins,” is commonly known and is not reflective of any unique features of the materials prepared in this study. Why did the authors not

compare samples with intergrowths to pure SAPO-34 and SAPO-18? In the end, the only new information presented in this section is the coking analysis, which is quite interesting. To this end, the authors need to justify the need for including the MTO results.

Reply: Thank you for your comments. In this work, we are focused on discussing how the diffusion of reactants and products affects the MTO process in different macro-architectures, and we use the catalytic test to illustrate the effectiveness of structural modification as a supplementary result. To better reflect the improvement of modified catalysts in this work, we added the data of micron-sized cuboid sample in Fig. 5a and b (as shown in Fig. R8) as a control group, which can be considered as solid cuboid formed by nearly pure SAPO-34 lattice. As shown in Fig. R8, the conversion rate of methanol in the micron-sized solid cuboid catalysts decreased rapidly after 90 min, while the modified catalysts could maintain a high conversion rate (>95%) for over 130 min. Combined with the imaging of hydrocarbons (cokes) inside cages, we can better understand how the blocked diffusion by coking limits the lifetimes of catalysts based on a Go-inspired Ising model. Different catalytic performances can be explained consistently by the different sizes of solid areas, that is, the different diffusion distances (micron-sized cuboid>hierarchical>sandwich) in three catalysts. Thus, the object of comparison in Fig. 5 should be the catalysts with different macro-architectures, not just pure SAPO-34 or SAPO-18. Since we choose the catalysts with a same Si/Al ratio in framework, the catalytic performance will not be affected by the microstructure (cages) itself, and mainly be affected by the macro-architecture (of course, it is probably related to microstructure). That is why we used a micron-sized solid cuboid particle as a control group to reflect the improvement of catalyst lifetime in Fig. 5.

Based on the above discussions, we think it is necessary to add the MTO results in the revised manuscript after we add the data of micron-sized cuboid sample. Since the highlights of this paper are the modification and the ultrahigh resolution analysis of catalyst structures, the catalytic test is an important and necessary supplementary result to check if these modified structures are effective in the catalytic applications. More importantly, based on the iDPC-STEM, we imaged the hydrocarbon pools and cokes inside cages for the first time, which agrees with the discrete Ising model that we have established previously. The coking analysis explains the deactivation mechanism related to catalyst structures. Thus, structure analysis, catalytic results and deactivation mechanism are unified in these discussions.

7) Minor Comments:

On page 5 the authors refer twice to “Fig. 3g and h”, but it seems this should be in reference to Figure 2.

In Fig. 2g and h, how do the authors know that atoms labelled Al and/or P cannot also be Si?

In the caption for Fig. 4c, it is stated “crystal in Fig. 3” but there are two crystals shown in that figure. The authors should be more specific.

In the caption of Fig. 5, the line starting with “e-g” should be “e-h”.

Reply: Thank you for your reminder. We have corrected all these mistakes in the revised manuscript. We can distinguish between the Al and P atom columns because the contrast difference due to the Si substitution can be ignored in the images. The Si content in zeolite framework is quite low in two SAPO-18/34 intergrowths (about 0.1). Such low Si content cannot change the relative height of the Al and P peaks in the profile analysis. We can still identify these two element columns by comparing the peak heights qualitatively.

Referee #3

Thank you for your time and efforts on our manuscript. We are glad to see your positive comments on the novelty and significance of our electron microscopic characterization and atomic structural analysis. In this work, we reveal the atomic structures of zeolite intergrowths by identifying light atoms and bonds, and control the synthesis of zeolite catalysts with short diffusion distances by adjusting their micro-structures. The catalytic performances illustrate the effectiveness of structural modification as a supplementary result, which provides valid ideas for the improvement of zeolite catalysts. These results will attract general interests for a broad readership in the atomic insights into not only the porous catalysts, but also other beam-sensitive and light-element materials. Based on your suggestions, we have supplied a lot of macroscopic characterizations to our manuscript, and revised the discussions on catalytic performances. Please see details in the following replies to your comments point to point. We will appreciate your positive suggestions on further improving this manuscript.

1. This study will be of interest to a narrow audience, which shouldn't be the case of an article of a journal that publishes results from a very broad range of topics.

Reply: The authors cannot agree with this statement. This work exhibits at least following breakthroughs in the microscopy, catalysis and material science. First, the iDPC-STEM greatly improves the level of electron microscopy for beam-sensitive materials. Based on the iDPC-STEM technique, we achieve the atomic imaging of silicoaluminophosphate zeolites for the first time. The Al, O, P atoms and Al-O, P-O bonds can be clearly identified with ultra-high resolution and signal-to-noise ratio. To the best of our knowledge, no other research group has achieved such imaging quality.

Then, such progresses on imaging technique can provide an atomic insight into the structures of intergrowth catalysts. We precisely control the mixture and separation of SAPO-34 and SAPO-18 lattices by simply changing the synthesis conditions (including the Si contents and templates). We reveal how such differences in micro-structures induce the totally different macro-architectures of catalyst crystals, since the sizes of solid areas (diffusion distances of products) deeply influence the gas selectivity and lifetime of catalysts. More importantly, we image the hydrocarbon pools and cokes inside cages by the iDPC-STEM, which agrees with the discrete Ising model that we established before and explains the deactivation mechanism related to catalyst structures.

Finally, these results enlighten the readers to recognize the unique advantages of iDPC-STEM technique for other beam-sensitive and light-element materials, such as MOFs, 2D materials, polymers and more inorganic/organic hybrid systems. Such structure analysis will not only bring us new understandings on different catalyst systems based on these materials, but also help us to reveal other novel physical and chemical properties of these materials. Thus, we believe that this work is of general interest for a broad readership of Nature Communications.

2. The relationship found by the authors between the synthesis conditions and the formation of different intergrowths is interesting, but it is not evident that it will lead to the discovery of new significantly better catalysts.

Reply: In this work, we are focused on discussing how the diffusion distances of gas reactants and products affects the MTO process in different macro-architectures, and we use the catalytic test to illustrate the effectiveness of structural modification as a supplementary result. To better reflect the improvement of modified catalysts in this work, we added the data of micron-sized cuboid sample in Fig. 5a and b (as shown in Fig. R1) as a control group, which can be considered as solid cuboid. As shown in Fig. R1, the conversion rate of methanol in the micron-sized solid cuboid catalysts decreased rapidly after 90 min, while the modified catalysts can maintain a high conversion rate (>95%) for over 130 min. Combined with the imaging of hydrocarbons (cokes) inside cages, we can understand how the blocked diffusion by coking limits the lifetimes of catalysts based on a Go-inspired Ising model. We achieved such improvement by modifying the macrostructures of SAPO-34/SAPO-18 intergrowths using different synthesis conditions (simply changing the Si content and templates). The hierarchical pores and lamellas in the modified catalysts reduce the diffusion distances of gas reactants and products. Therefore, different catalytic performances can be explained consistently by the different sizes of solid areas, that is, the different diffusion distances (micron-sized cuboid>hierarchical>sandwich) in three catalysts. These results prove our strategy to control the synthesis of SAPO-34/SAPO-18 with hierarchical pores and lamellas to reduce the diffusion distances of gas reactants and products, which will lead to the discovery of better catalysts with high conversion, high selectivity and long lifetime.

Moreover, exploring the basic correlation between structures and properties is the most important subject in materials science. Clear structural characterization is a prerequisite for the discovery, design and modification of new materials with new functions. As for the zeolites (such as SAPO-34/18 intergrowths), atomic structures are still challenging to be resolved due to the inevitable damages by high-energy electrons and low contrasts of light elements. It is a major bottleneck for the studies of zeolite catalysts, which have been widely used in chemical industry for several decades. In this work, we confirm that the iDPC-STEM can be applied in the atomic imaging of zeolite frameworks, and it is used to reveal how different architectures of catalysts are generated to improve the catalytic performances based on different local structures. Meanwhile, such iDPC-STEM imaging can be extended to other catalysts based on various porous materials. There are over 250 different types of zeolites and their intergrowth families (<http://www.iza-structure.org/databases/>). They have been used in various important chemical processes, and most of them have not been atomically resolved by electron microscopy. Moreover, MOFs and COFs have also been studied as porous catalysts and supporters for other catalysts, which cannot be atomically imaged due to their beam-sensitive nature. The results in this work will

enlighten readers to recognize the unique advantages of iDPC-STEM for imaging all these beam-sensitive light-element catalysts, and bring new understandings and discoveries in these catalyst systems.

Figure R1. Catalytic performances of SAPO-34/18 catalysts. **a**, Conversion of methanol over three SAPO-34/18 catalysts at different times on stream. The error bars represent the standard deviations of data in repeated experiments. **b**, Cumulative gas selectivity of the main products in the methanol conversion over three SAPO-34/18 catalysts, including the light olefins, light alkanes and heavy hydrocarbons. **c-h**, Gas selectivity of the main products during the methanol conversion. These catalysts show very high gas selectivity of light olefins (C₂-C₄ olefins, ~93%), while a few alkanes (C₁-C₄ alkanes, ~2%) and heavy hydrocarbons (C₅-C₇, ~5%) were produced.

- The improvements in the catalytic conversion of methanol described in the final part of the manuscript are minor and do not constitute a significant breakthrough; although, they support the claims of the authors.

Reply: As shown in Fig. R1, we can observe the improvement of the performances

of our modified catalysts, especially on the lifetime and ethylene to propylene ratio. The highlights of this work are the high-quality imaging, atomic structural analysis, controllable catalyst structures and how they affect the catalytic performances. The role of catalytic tests is to illustrate the effectiveness of structural modification as a supplementary result, not to discover a new catalyst. In the catalytic tests, we only used pure zeolite catalysts in order to study the effects of gas diffusion on MTO process in a clean and clear system, since only the diffusion distances are different in these catalysts significantly. As we expected, these catalytic results support our strategy of catalyst modification and observed local structures of different SAPO-34/18 intergrowths, which just can satisfy our requirements and agree with the main conclusions in this manuscript. Meanwhile, based on our catalytic tests of three catalysts, the conversion rate of methanol is near 100% at the early stage and the selectivity of light olefins is over 90%. Such performances are good enough in previous studies on similar SAPO-34/18 catalysts. The lifetimes of catalysts are related to test conditions (temperature, WHSV and reactor type), which cannot be simply compared between the results in references. Thus, on the one hand, we think that the catalytic performances are good enough and strongly support our strategy and characterizations. On the other hand, it is one-sided to judge the novelty and significance of this work only by these catalytic results, since the main highlights are not only these catalytic applications.

4. The different materials prepared by modifying the synthesis conditions lack a proper characterization, beyond the excellent microscopy work described in the manuscript.

Reply: The authors agree with this point, but the required characterizations can be easily supplied in the revised manuscript, which should not be a reason for rejection. In this work, we at least provide the following characterizations on SAPO-34/18 catalysts beyond the beautiful iDPC-STEM imaging. SEM imaging (Fig. R2) can also reveal the surface morphologies and 3D structures of different catalysts. PXRD results (Fig. R3) give the periodic information of SAPO-34 and SAPO-18 lattices, which can be used to estimate the contents of SAPO-34 and SAPO-18 zeolites in intergrowths. EDS mapping (Fig. 3 in the revised manuscript) provides the spatial distribution of elements in catalysts, while the XRF helps us determine the overall Si/Al ratios in frameworks. Meanwhile, to better analyze the porous structures as you requested, we added the adsorption isotherms of different intergrowth catalysts (Fig. R4) in supplementary information. The specific surface areas of three catalysts can be calculated based on the BET equation, which are 694.03 (sandwich), 638.5 (hierarchical) and 613.4 (micro-cuboid) m²/g respectively (mainly contributed by micropores). We can observe the hysteresis loops in the isotherms of the sandwich and hierarchical catalysts, which indicates the mesopores in these two catalysts (the mesopores in sandwich samples are mainly caused by slits between lamellas), while there is no hysteresis in

micron-sized cuboid sample. Especially, for the hierarchical catalyst, there are different types of pores inside, including micropores, mesopores, macropores and hollow areas (not only mesopores, we have revised the statement in the main manuscript). We used the Hg intrusion method to reveal the pore size distribution in three catalysts (Fig. R5). It is obvious to detect the mesopores and macropores with the pore sizes ranging from 10 to 1000 nm in the sandwich and hierarchical catalysts, while there is nearly no mesopores and macropores in the micro-cuboid catalyst. We have added all these characterizations into the revised manuscript and supplementary information.

Figure R2. SEM images of three intergrowth catalysts.

Figure R3. PXRD results of three intergrowth catalysts.

Figure R4. Adsorption isotherms of three intergrowth catalysts.

Figure R5. Hg intrusion results of three intergrowth catalysts.

Reviewers' Comments:

Reviewer #1:

Remarks to the Author:

Dear authors,

I am pleased to see that you have addressed my remarks and properly responded to my comments and concerns. Therefore, I now recommend your manuscript for publication.

One last simple correction:

1) In the line 60 of your manuscript please remove the word "theoretically". The technique has been proven both theoretically and practically numerous times by now.

Best regards

Reviewer #2:

Remarks to the Author:

Review is attached.

Review, NCOMMS-20-32495A

In the original review we had expressed the following two opinions: (i) the TEM work is truly beautiful and captures aspects of intergrowths that have not been previously reported; and (ii) the roles of TEA and TEOH in directing intergrowths are not definitively proven, the impact of Si doping is ambiguous, and the catalysis experiments lack clear conclusions (i.e. in relation to structure-performance relationships).

The authors have done an excellent job addressing the comments and concerns raised in the original review. Each of these comments is listed below along with a response to indicate how they were addressed in this revised version of the manuscript. In summary, the authors have adequately addressed each point and it is recommended that this revised version of the manuscript be accepted without requests for additional edits.

Details of the Original Review Comments and Responses:

Comment 1: On page 3 the authors make the following statement: “two kinds of SAPO-34/18 intergrowth catalysts were successfully synthesized by changing the templates and the contents of reactants.” It is not clear how TEA and TEOH direct different intergrowths. How many total experiments were performed? How reproducible are the syntheses? It wasn't clear if the authors used pure TEOH, and in the case of mixtures of TEA and TEOH how many different combinations were tested? Did the authors see a progressive shift in intergrowths from pure TEA to pure TEOH?

Response: the authors have nicely addressed this point and included two additional figures (Fig. 3 and Fig. S9). No additional edits are requested.

Comment 2. Mechanistic arguments made in Figure 3 are unclear. The random orientations shown in Fig. 3b appear similar to those in Fig. 3c and d, which seems to indicate that mesopore formation cannot be linked to the frequency of lattice mismatching. The authors need to provide more definitive evidence of characteristic features in the crystal structure that may lead to mesopore formation. Generalizing these trends based on limited examples is also problematic.

Response: the authors have explained this in greater detail, and the images in Figure R3 within the rebuttal letter are also very convincing. No additional edits are requested.

Comment 3. The reviewer strongly disagrees with the statement on page 8: “the Si content of the SAPO-34 domain is significantly higher than that of the SAPO-18 domain, since the Si/Al ratio rises from ~0.05 to ~0.2 through the SAPO-18 to SAPO-34 interface.” The crystal used for EDS analysis has an odd shape. The center of the crystal appears to be much lower (i.e. thinner) than the two ends. This likely gives rise to the drop in elemental intensity in the center region. All elements have lower intensity, and it is likely the resolution for Si (the minor component) under these conditions is subject to error, giving rise to the lower Si/Al ration in Fig. 4b. Do the authors have examples of crystals that are level? The crystal in Fig. 3e is odd and seems to be an abnormality (unless the authors have shown this is representative of many crystals prepared in the synthesis batch. Also, what are the Si/Al ratios of pure SAPO-34 and SAPO-18 (these materials are not mentioned)? Does one traditionally have a different Si/Al ratio? This mechanistic

argument requires more evidence – particularly if the authors are going to make the following definitive statement in the conclusion section: “the enrichment of Si element is directly related to the switch between the SAPO-34 and SAPO-18 lattices.”

Response: the authors have provided a convincing case along with additional analyses to support their claim. As such, the original concern has been addressed. No additional edits are requested.

Comments 4-6. There are several comments and concerns with the catalysis section...

Responses: the authors have done a very nice job of addressing these points by making changes to their figures and adding additional data. This is very much appreciated and has improved the clarity of the manuscript. No additional edits are requested.

Minor Comments: the authors have addressed all of these comments. No further changes are needed.

Reviewer #3:

Remarks to the Author:

The manuscript has been greatly improved by addressing the comments raised by the reviewers, which - from different perspectives - have significantly contributed to include aspects that were not sufficiently addressed in the original version.

More specifically, the additional characterization of these materials, including SEM (BTY, the scale bars are missing), PXRD, EDS mapping (in Figure 4 not 3), Hg porosimetry, and specially the Ar adsorption/desorption isotherms at 87 K are very useful as they provide a more complete picture of the morphology, composition, and textural properties of these materials. Based on both the Hg porosimetry and Ar isotherms, it seems that the pores would be better characterized as macropores (with maybe a small contribution of mesopores). This is quite clear by Hg porosimetry, and by the very flat plateau of the Ar isotherms until $P/P_0 > 0.9$ also indicate rather large pores.

Regarding the catalysis section, I agree that these results are only to illustrate the (well-known) role of hierarchical porosity in the MTO process.

The additional results – specially the more complete characterization of the materials -, the corrections, and clarifications made by the authors are satisfactory.

Reviewer #1

Comments:

I am pleased to see that you have addressed my remarks and properly responded to my comments and concerns. Therefore, I now recommend your manuscript for publication.

One last simple correction:

1) In the line 60 of your manuscript please remove the word “theoretically”. The technique has been proven both theoretically and practically numerous times by now.

Reply: Thank you for your recommendation. We have removed the word “theoretically”.

Reviewer #2

Comments:

In the original review we had expressed the following two opinions: (i) the TEM work is truly beautiful and captures aspects of intergrowths that have not been previously reported; and (ii) the roles of TEA and TEAOH in directing intergrowths are not definitively proven, the impact of Si doping is ambiguous, and the catalysis experiments lack clear conclusions (i.e. in relation to structure-performance relationships).

The authors have done an excellent job addressing the comments and concerns raised in the original review. Each of these comments is listed below along with a response to indicate how they were addressed in this revised version of the manuscript. In summary, the authors have adequately addressed each point and it is recommended that this revised version of the manuscript be accepted without requests for additional edits.

Reply: Thank you for your recommendation. We are very glad and encouraged that our responses addressed all your comments. Based on your detailed responses, we find that no additional edits are requested, so there is no new comment that need to be addressed now.

Reviewer #3

Comments:

The manuscript has been greatly improved by addressing the comments raised by the reviewers, which - from different perspectives - have significantly contributed to include aspects that were not sufficiently addressed in the original version.

More specifically, the additional characterization of these materials, including SEM (BTY, the scale bars are missing), PXRD, EDS mapping (in Figure 4 not 3), Hg porosimetry, and specially the Ar adsorption/desorption isotherms at 87 K are very useful as they provide a more complete picture of the morphology, composition, and textural properties of these materials. Based on both the Hg porosimetry and Ar isotherms, it seems that the pores would be better characterized as macropores (with maybe a small contribution of mesopores). This is quite clear by Hg porosimetry, and by the very flat plateau of the Ar isotherms until $P/P_0 > 0.9$ also indicate rather large pores.

Regarding the catalysis section, I agree that these results are only to illustrate the (well-known) role of hierarchical porosity in the MTO process.

The additional results – specially the more complete characterization of the materials -, the corrections, and clarifications made by the authors are satisfactory.

Reply: Thank you for your recommendation. We are very glad and encouraged that you are satisfied with our responses. Based on your detailed comments, (1) the missing scale bars are added in the SEM images in Fig. 1; (2) the figures in the response file have been moved into the main manuscript or SI; (3) we agree that both the macropores and mesopores contribute to the pore structures in SAPO-34/18 (so called the hierarchical catalyst) and we have emphasized this point in the manuscript.